# Rho GTPase activity crosstalk mediated by Arhgef11 and Arhgef12 coordinates cell protrusion-retraction cycles

Suchet Nanda [1,2,5], Abram Calderon[1,2,5], Arya Sachan [1,5],
Thanh-Thuy Duong[1,2,5], Johannes Koch[3], Xiaoyi Xin[4],
Djamschid Solouk-Stahlberg[1,2], Yao-Wen Wu [4], Perihan Nalbant [3] ✉ &
Leif Dehmelt [1] ✉

Rho GTPases play a key role in the spatio-temporal coordination of cytoskeletal dynamics during cell migration. Here, we directly investigate crosstalk between the major Rho GTPases Rho, Rac and Cdc42 by combining rapid activity perturbation with activity measurements in mammalian cells. These studies reveal that Rac stimulates Rho activity. Direct measurement of spatio-temporal activity patterns show that Rac activity is tightly and precisely coupled to local cell protrusions, followed by Rho activation during retraction. Furthermore, we find that the Rho-activating Lbc-type GEFs Arhgef11 and Arhgef12 are enriched at transient cell protrusions and retractions and recruited to the plasma membrane by active Rac. In addition, their depletion reduces activity crosstalk, cell protrusion-retraction dynamics and migration distance and increases migration directionality. Thus, our study shows that Arhgef11 and Arhgef12 facilitate exploratory cell migration by coordinating cell protrusion and retraction by coupling the activity of the associated regulators Rac and Rho.

Cytoskeletal dynamics drive force-generating mechanisms that control changes in cell shape during cell morphogenesis and cell migration[1]. These force-generating mechanisms, in turn, are controlled by signaling networks that are regulated in space and time[2]. Members of the Rho GTPase family play important roles in this process. Protrusive forces at the leading edge of cells are typically induced by the Rho GTPase family members Rac1 and Cdc42, which promote nucleation and polymerization of actin filaments and associated proteins[3]. Rac1 is best known for its role in inducing flat cell protrusions, called lamellipodia, while Cdc42 preferentially induces pointed protrusions, called filopodia. RhoA, a related Rho GTPase family member, induces stress fibers that generate contractile forces that

originate from Myosin 2 motor mini-filaments that act on anti-parallel actin filaments[3].

Crosstalk between Rho GTPases appears to play an important role in the coordination of their activities in space and time[4]. In a simple concept, mutually inhibitory crosstalk between Rac1/Cdc42 and RhoA was proposed to spatially segregate cell protrusion and cell contraction between the protrusive, leading edge and the contractile, trailing rear, respectively[4]. Studies using Rho activity sensors in small migrating cells, such as neutrophils, have produced results that are consistent with this idea[5,6]. However, in larger migrating cells, such as fibroblasts, protrusive Rac1/Cdc42 and contractile RhoA signals were both detected near the leading edge of migrating cells, suggesting a more

[1]Fakultät für Chemie und Chemische Biologie, TU Dortmund University, 44227 Dortmund, Germany. [2]Department of Systemic Cell Biology, Max Planck Institute of Molecular Physiology, 44227 Dortmund, Germany. [3]Department of Molecular Cell Biology, Center of Medical Biotechnology, University of Duisburg-Essen, 45141 Essen, Germany. [4]SciLifeLab and Department of Chemistry, Umeå Centre for Microbial Research, Umeå University, 90187 Umeå, Sweden. [5]These authors contributed equally: Suchet Nanda, Abram Calderon, Arya Sachan, Thanh-Thuy Duong.
✉e-mail: perihan.nalbant@uni-due.de; leif.dehmelt@tu-dortmund.de

complex relationship between these Rho GTPases[7-10]. Furthermore, mesenchymal cells typically generate highly dynamic regular or irregular cycles of cell protrusion and retraction near the leading cell edge, which are thought to play an exploratory role in cell migration[11]. Thus, in such cells protrusion and retraction dynamics appear to be tightly coupled in space and time. Therefore, specific mechanisms must exist that generate these highly dynamic cycles of cell protrusion and retraction.

Recent studies revealed transient activity pulses or traveling waves of signal network components that promote cell protrusion, including Cdc42, actin nucleation promoting factors (NPFs), and Ras-type GTPases[6,12-14]. These activity patterns are typical for signal networks that combine both positive and negative feedback regulation to generate excitable or oscillatory system dynamics[15]. Furthermore, we and others recently found that the contractility-inducing GTPase Rho is a central component of a signal network that generates pulses and waves of cell contraction[16-18].

Here we hypothesize, that a link between these excitable or oscillatory signal networks might play a role in coordinating the highly dynamic cycles and bursts of cell protrusion and cell retraction that are observed in exploratory, mesenchymal cell migration. Previous observations of Rho GTPase crosstalk have relied on slow perturbations and/or indirect evidence. Here, we combine rapid activity perturbation techniques based on chemically-induced dimerization (CID) as well as optogenetic approaches with continuous monitoring of response dynamics to directly investigate Rho GTPase crosstalk in living cells. Importantly, the perturbation constructs that were used in our studies are based on constitutively active Rho GTPase mutants[19,20], which are not prone to potential feedback regulation and adaptive responses. We thereby minimize secondary effects, which might otherwise mask cause and effect relationships. As expected from their proposed mutual inhibition, activation of RhoA leads to a decrease in Rac activity. Surprisingly, our studies revealed the activation of Rho after Rac1 activation, which is contrary to the expected mutual inhibition between these signals. Furthermore, by using improved sensors to measure endogenous Rac1 and Rho activity, we reveal a tight correlation between increased, local Rac activity at the plasma membrane during cell protrusion and increased Rho activity during cell retraction. Investigations into the molecular mechanisms reveal that the Rho-activating Lbc-type GEFs Arhgef11 and Arhgef12 act as Rac effectors and thereby can mediate the observed Rac/Rho activity crosstalk. Furthermore, our data shows that Arhgef11 and Arhgef12 are required for effective spatio-temporal coupling between cell protrusion and retraction dynamics, and that this is critical for efficient exploratory cell migration.

## Results

### Analysis of Rho GTPase crosstalk in living cells

To directly investigate Rho GTPase crosstalk, we sought to develop methods that enable the combination of rapid activity perturbations and simultaneous activity monitoring in individual, living cells. To induce such rapid perturbations, we extended an approach that we developed previously[19], which is based on reversible, chemically-induced plasma membrane targeting of constitutively active Rho GTPase mutants (Fig. 1a). Analogous to the published approach[19], we removed the C-terminal CAAX motif found in Rho GTPases, which normally acts as a membrane anchor after post-translational geranylgeranylation[21-23]. Instead, we added the FKBP' heterodimerization domain. This allowed us to induce reversible targeting of Rho GTPases from the cytosol to the plasma membrane, where they interact with a plasma membrane-anchored heterodimerization partner (eDHFR) upon addition of a chemical dimerizer (SLF'-TMP). In our previous study, we found that the recruitment of constitutively active Rac1 induced reversible formation of lamellipodia, which are a typical phenotype of increased Rac1 activity[19]. Here, we extended this

perturbation strategy to the three best-characterized Rho GTPases Rac1, Cdc42, and RhoA. As Neuro-2a neuroblastoma cells show substantial morphological changes in response to active Rac1[24], we used this cell line for our initial analyses. As expected, Rac1 and Cdc42 plasma membrane targeting induced the formation of cell protrusions, while RhoA plasma membrane targeting induced cell contraction (Fig. 1b, Supplementary Fig. 1a and Supplementary Movie 1). Cdc42-induced protrusions were characterized by both filopodial and lamellipodial structures, which is in agreement with the known partial overlap in their induced phenotypes[25]. Inhibition of plasma membrane targeting with a small molecule competitor (TMP) reversed the Rho GTPase-induced phenotypes (Fig. 1b, Supplementary Fig. 1a and Supplementary Movie 1).

Next, we combined Rho GTPase perturbations with Rho GTPase activity measurements (Fig. 2a). To investigate the Rho, Cdc42, and Rac activity response, we used translocation-based activity sensors[17,26-28]. These sensors are based on the plasma membrane recruitment of GTPase-binding domains (GBDs) from specific effector proteins: Rhotekin[27], WASP[26,28], and p67[phox 17], respectively. In our specific implementation, we combined these domains with the very low expressing delCMV promoter and used TIRF microscopy for sensitive readout of plasma membrane translocation[17]. While these effector domains are known to be specific for the respective Rho, Cdc42, and Rac subgroups, they are not expected to distinguish between more closely related family members. The Rhotekin-based Rho sensor will detect active RhoA, RhoB, and RhoC, the WASP-based Cdc42 sensor will detect active Cdc42, TC10, and TCL and the p67[phox]-based Rac sensor will detect Rac1, Rac2, and Rac3.

The effectors and cellular functions of these closely related GTPases are very similar. Therefore, we consider their combined activity for our crosstalk analyses and refer to sensor measurements using the corresponding Rho GTPase subfamily names Rac, Cdc42, and Rho. During the time course of dimerizer-induced activity perturbations, significant changes in cell volume can occur (Supplementary Fig. 1c), which could also change the TIRF signal. To control for this potential artifact, we co-transfected a cytosolic cell volume marker that acts as a control construct and used this to correct changes in the sensor signal (see "Methods"). Figure 2b, c shows the Rac1 perturbation and the raw, uncorrected Rho activity sensor and control sensor response measurements in a representative cell (see also Supplementary Movie 2). Here, an increase of the Rho sensor response was accompanied by a decrease in the control sensor signal, showing that the uncorrected sensor measurement underestimated the actual response.

Figure 2d and Supplementary Fig. S2a show the kinetics for all Rho GTPase crosstalk combinations. The perturbation in the majority of individual cells follows biphasic kinetics, which might be caused by simultaneously occurring processes of plasma membrane targeting and intracellular diffusion of the GTPases and uptake of the small molecule dimerizer through the plasma membrane. The Cdc42 perturbation was considerably weaker compared to the Rac1 or RhoA perturbation, however, the corresponding Cdc42 sensor response was particularly high, suggesting that the corresponding perturbation was nevertheless very effective (Supplementary Fig. 2a, b).

Several crosstalk measurements were in agreement with previous observations. As expected, the strongest response in the activity for a particular sensor was observed upon activation of the corresponding Rho GTPase (Supplementary Fig. 2a, b). This observation further supports the specificity of the effector domains for the individual Rho family GTPases. In a previous study[17], we observed that Rho activity pulses precede increased Cdc42 activity with a delay of $10.1 \pm 1.8$ s, suggesting that Cdc42 might be activated downstream of Rho activity. However, these previous observations only indicated a correlation between Rho and Cdc42 and did not show a causal relationship between these activities. Here we found that activation of RhoA indeed

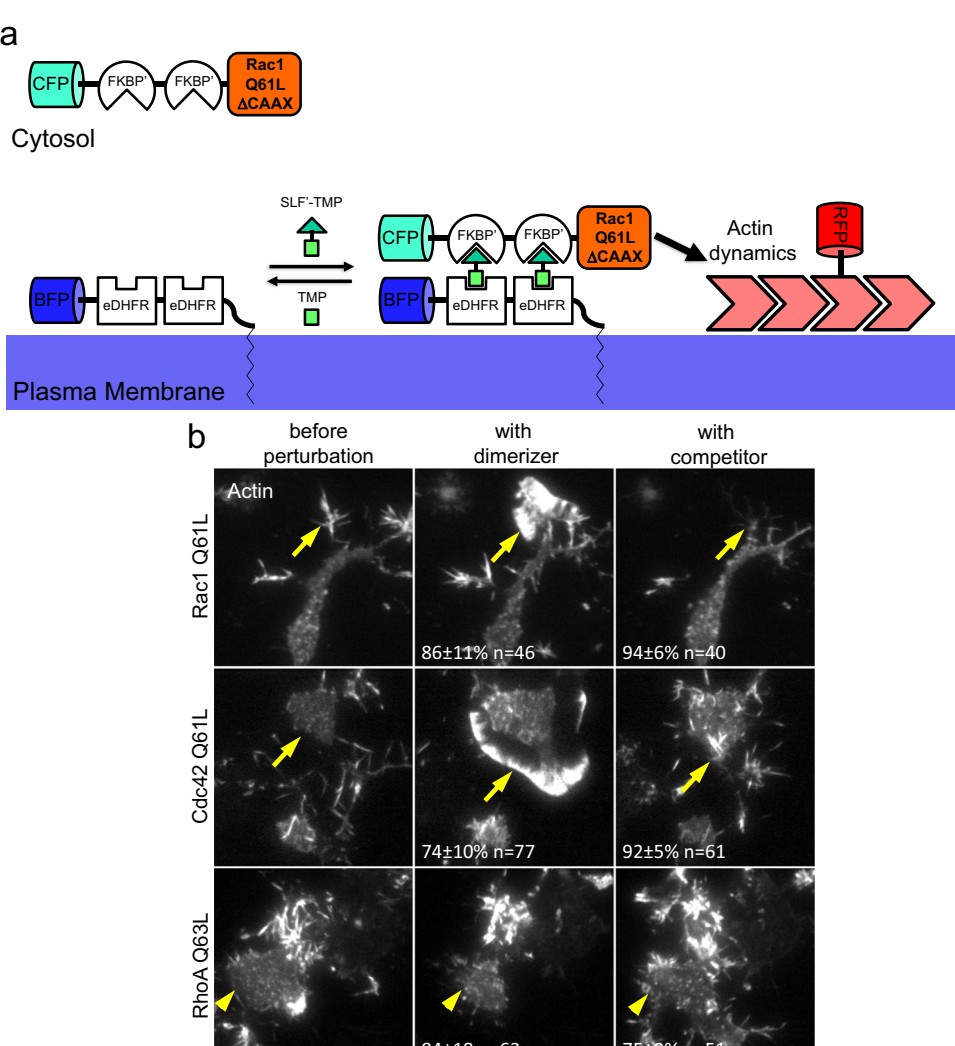

**Fig. 1 | A general method for rapid perturbation of Rho GTPase activity in living cells. a** Schematic of rapid, reversible Rho GTPase activity perturbation via chemically-induced dimerization and readout of cell morphology and cytoskeletal organization by an actin-reporter. **b** Representative frames from TIRF microscopy time series of mCherry-Actin obtained 30 s before, 24 min during and 24 min after Rho GTPase activation in Neuro-2a neuroblastoma cells (see also Supplementary Movie 1). Yellow arrows point to cell areas that reversibly generate protrusions during Rac1 or Cdc42 activation, and yellow arrowheads point to areas that undergo reversible retraction during RhoA activation. Observations are representative for 3 independent repetitions with a total of at least 40 cells per condition (exact numbers of cells are indicated in individual panels). Numbers in middle panels indicate percentage ± standard error of the mean of cells that initiate protrusion (Rac1/Cdc42) or retraction (RhoA) after addition of dimerizer. Numbers in panels indicate percentage of reacting cells that showed a phenotypic reversal. Scale bar: 10 μm; 0.26 μm/pixel; CFP cyan fluorescent protein, BFP blue fluorescent protein, RFP red fluorescent protein, FKBP' FK506-binding protein with F36V mutation, eDHFR *E. coli* dihydrofolate reductase, SLF' synthetic ligand of FKBP', TMP eDHFR interacting small molecule trimethoprim.

stimulates Cdc42 activity (Fig. 2d, e). We also observed inhibition of Rac activity by RhoA activation, which is in agreement with the previously proposed mutual inhibition between these GTPases (Fig. 2d, e)[4,29–32]. Interestingly, in our studies, activation of Rho was the strongest and most robust crosstalk after activation of Rac (Fig. 2d, e and Supplementary Movie 2). This was surprising as it contradicts the proposed mutual inhibitory relationship between RhoA and Rac1[4]. Finally, we observed significant activation of Cdc42 by Rac1, which could result either from direct crosstalk between these GTPases, or from indirect activation via Rho (Fig. 2d, e, f).

As the observation that Rac1 activates Rho was unexpected, we sought to further investigate this relationship with an alternative method. Therefore, we used the light-controlled PA-Rac1 construct which was previously developed by fusing a LOV2 domain to constitutively active Rac1[20]. After illumination with a wavelength of 445 nm, this LOV2-domain quickly unfolds within milliseconds and unblocks active Rac1, which can then interact with its effectors[20]. In the absence of 445 nm light, the LOV2 domain refolds within several seconds, which again blocks the associated active Rac1[17,20,33]. By combining this method with Rho activity measurements (Fig. 3a), we confirmed Rac1-dependent Rho activation in Neuro-2a cells (Fig. 3b, Supplementary Fig. 3a, b). In contrast, very little change in signal was observed using the control sensor (Fig. 3b, Supplementary Fig. 3a, b).

We next sought to determine whether Rac1-induced Rho activation was unique to Neuro-2a neuroblastoma cells. Therefore, we repeated the photoactivation experiment in a panel of commonly used cell lines, including NIH3T3, HeLa, U2OS and A431 cells. Similar as in Neuro-2a cells, we observed a significant activation of Rho during continuous Rac1 activation in these cells (Fig.3b, Supplementary Fig. 3a, b). However, while Neuro-2a, NIH3T3 and HeLa cells predominantly responded with a constant, increased Rho activity level, U2OS (Fig. 3b, Supplementary Fig. 3a, b) and A431 cells (Fig. 3c–e), responded more transiently and dynamically. This more dynamic response suggests additional regulatory mechanisms. In particular, the

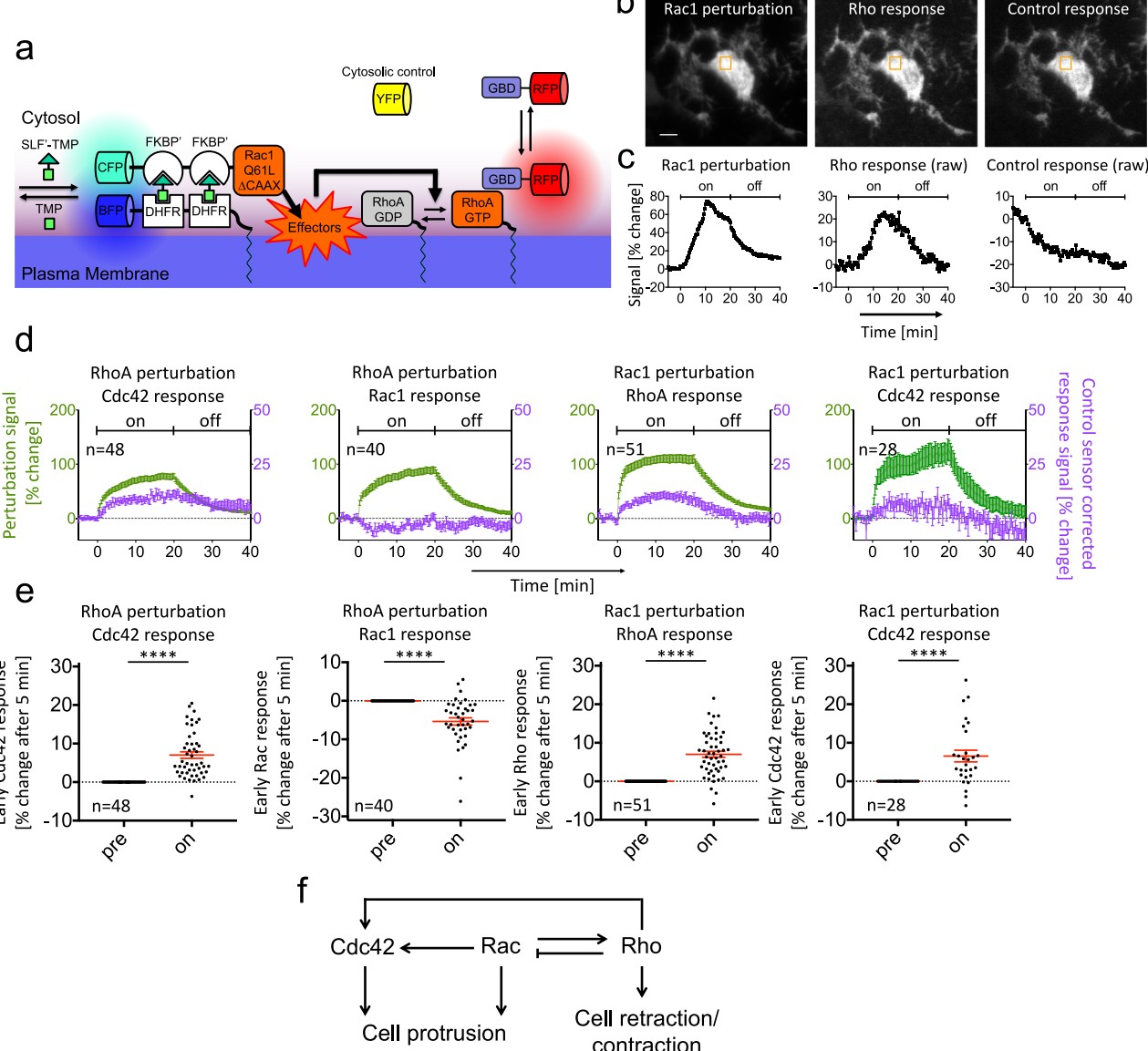

**Fig. 2 | Analysis of Rho GTPase crosstalk in living cells. a** Schematic of rapid activity perturbation and combined activity measurement strategy. **b−e** Analysis of perturbation-response relationships of Rho GTPase activity in Neuro-2a cells. **b, c** Representative TIRF images before dimerizer addition (**b**, top) and Rac1 perturbation and uncorrected, raw Rho sensor and raw control sensor signal kinetics (**c**, bottom) corresponding to orange boxes (see also Supplementary Movie 2). All constructs are predominantly cytosolic and homogenously distributed in the cell bodies and neurite-like protrusions. **d** Average perturbation and control-corrected activity sensor signal kinetics for selected crosstalk combinations (see Supplementary Fig. 2 for all combinations). **e** Quantification of average sensor signal

changes during Rho GTPase activity perturbation. Responses at time points before (pre), and 5 min after dimerizer addition (on) are shown. **f** Influence diagram that summarizes significant activity response measurements at 5 min after dimerizer addition. All observations and measurements are based on at least 3 independent repetitions with a total of at least 28 cells per condition (exact numbers of cells are indicated in individual panels). Scale bars: 5 μm; 0.26 μm/pixel; ****$P < 0.0001$; Student's $t$-Test. Error bars represent standard error of the mean. YFP yellow fluorescent protein, GBD GTPase-binding domain, All statistical tests were two-sided. Source data are provided as a Source Data file.

signal decrease that is observed with a short temporal delay in a sub-population of A431 cells (Fig. 3c−e) points to negative feedback regulation, which for example could be mediated by RhoA-dependent Rac inhibition[4]. Further characterization showed that the Rho activity response was dose-dependent (Supplementary Fig. 3d) and homogenous within the cell attachment area (Fig. 3g). This homogenous response corresponds to the similarly homogenous distribution of the PA-Rac1 perturbation construct in the plasma membrane.

**Rac and Rho dynamics in protrusion-retraction cycles**
The observed stimulation of transient Rho activity dynamics by Rac1 in U2OS and A431 cells might play a role in cellular processes that are

characterized by transient cell shape changes. We were particularly intrigued by the idea that the Rac1/Rho crosstalk might play a role in coupling these activities during cell protrusion-retraction cycles. A431 cells are well known to generate transient cell protrusions that are followed by transient cell retractions both after growth factor stimulation[34] and spontaneously[35].

To study a potential role of Rho GTPase crosstalk in this system, we first characterized the spatio-temporal coupling of Rac and Rho activity with cell shape changes. Based on well-established bio-chemical studies, the activity of Rac is expected to stimulate actin polymerization, leading to cell protrusion[36,37], and active Rho is expected to stimulate actomyosin contraction and cell retraction[38,39].

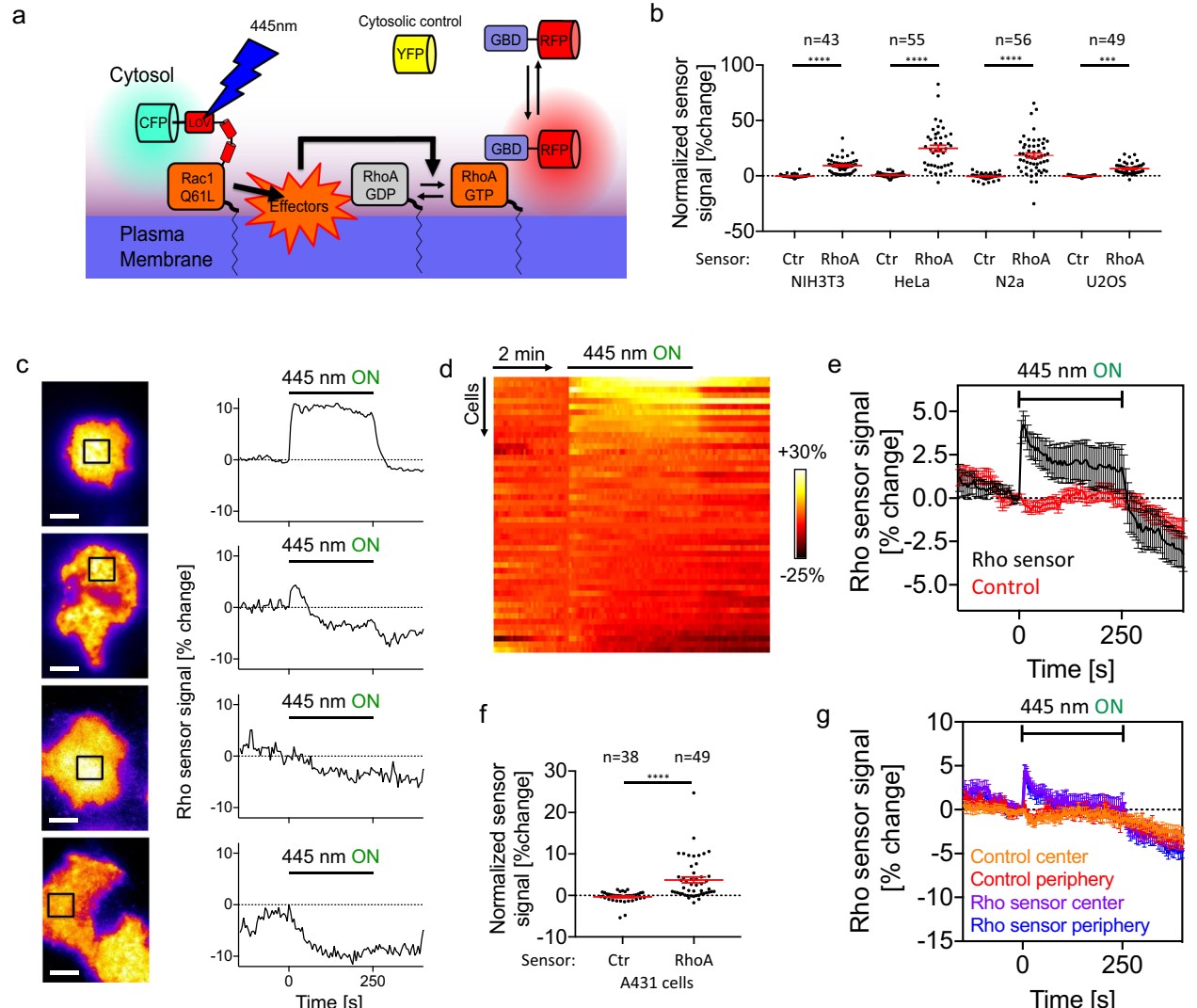

**Fig. 3 | Rac/Rho crosstalk is commonly observed in adherent mammalian cell lines and can trigger a dynamic Rho activity response. a** Scheme of optogenetic approach to measure Rac/Rho activity crosstalk. **b–f** Measurement of Rho activity dynamics during rapid optogenetic activation of Rac. **b, f** Rho activity response to Rac1 perturbation in the commonly used cell lines NIH3T3, HeLa, N2a and U2OS. The difference between 5 measurements before and 5 measurements after the onset of illumination is shown, each corresponding to a time frame of 25 s. **c** Typical Rho activity response dynamics observed in A431 cells (*n* = 49 cells from 3 independent experiments). Left: representative TIRF images from video microscopy time series. Right: Rho activity dynamics corresponding to black boxes in left panels. 67% of cells showed a discernible, positive, reversible Rho activity response. In particular, 29% showed a continuous Rho activation during Rac1 photoactivation

(top panels), 38% of the cells showed a single, transient activity pulse response and 25% showed no response (middle panels). 8% of cells showed a negative response (bottom panels). **d** Heatmap representation of all Rho activity responses measured in A431 cells. Color represents % change of the Rho activity sensor. **e** Measurement of average Rho activity sensor kinetics before, during and after Rac1 activation in A431 cells, corresponding to data shown in (**f**). **g** Measurement of average Rho activity sensor kinetics corresponding to data shown in (**e**) in peripheral or central cell attachment areas. Scale bars: 10 μm; 0.26 μm/pixel; ****$P < 0.0001$; ***$P < 0.001$; Student's *t*-Test. Error bars represent standard error of the mean. The time range of optogenetic activation is indicated in plots with the label "445 nm ON". All statistical tests were two-sided. Source data are provided as a Source Data file.

However, previous studies based on FRET biosensors suggested an opposite relationship, with active Rho being coupled to initial cell protrusion and subsequent Rac activation[8].

Initial experiments using the sensors described above showed only weak signals. This suggests that these sensors are not sensitive enough to detect the activation of Rho GTPases during spontaneous cell morphodynamics. To increase their sensitivity, we generated constructs that contain tandem GTPase-binding domains (GBDs) that could benefit from binding multiple GTPase molecules at the same time. Previous studies showed that this sensor design can improve sensitivity[40–42], presumably due to increased avidity. As these constructs can compete with endogenous effectors, we combined them with the very low expressing truncated delCMV promotor and

detected their localization using highly sensitive TIRF microscopy equipment. As shown in Supplementary Fig. 3e–g, signal changes measured using sensors that contain multiple GBDs were significantly higher compared to single GBD sensors after stimulation of Rac1 or Rho activity.

Using these improved sensors, we observed very tight spatio-temporal coupling between Rac activity and cell protrusion and Rho activity and cell retraction (Fig. 4a–e, Supplementary Fig. 4a–d see also Supplementary Movies 3, 4 and 5). To quantify the coupling between Rho GTPase activity and cell shape changes, we used the open-source ADAPT ImageJ plugin[43], which is well-suited to track cell edge movements and associated fluorescence signals at the cell edge (Fig. 4b). We first modified the plugin to optimize extraction of our sensor signals.

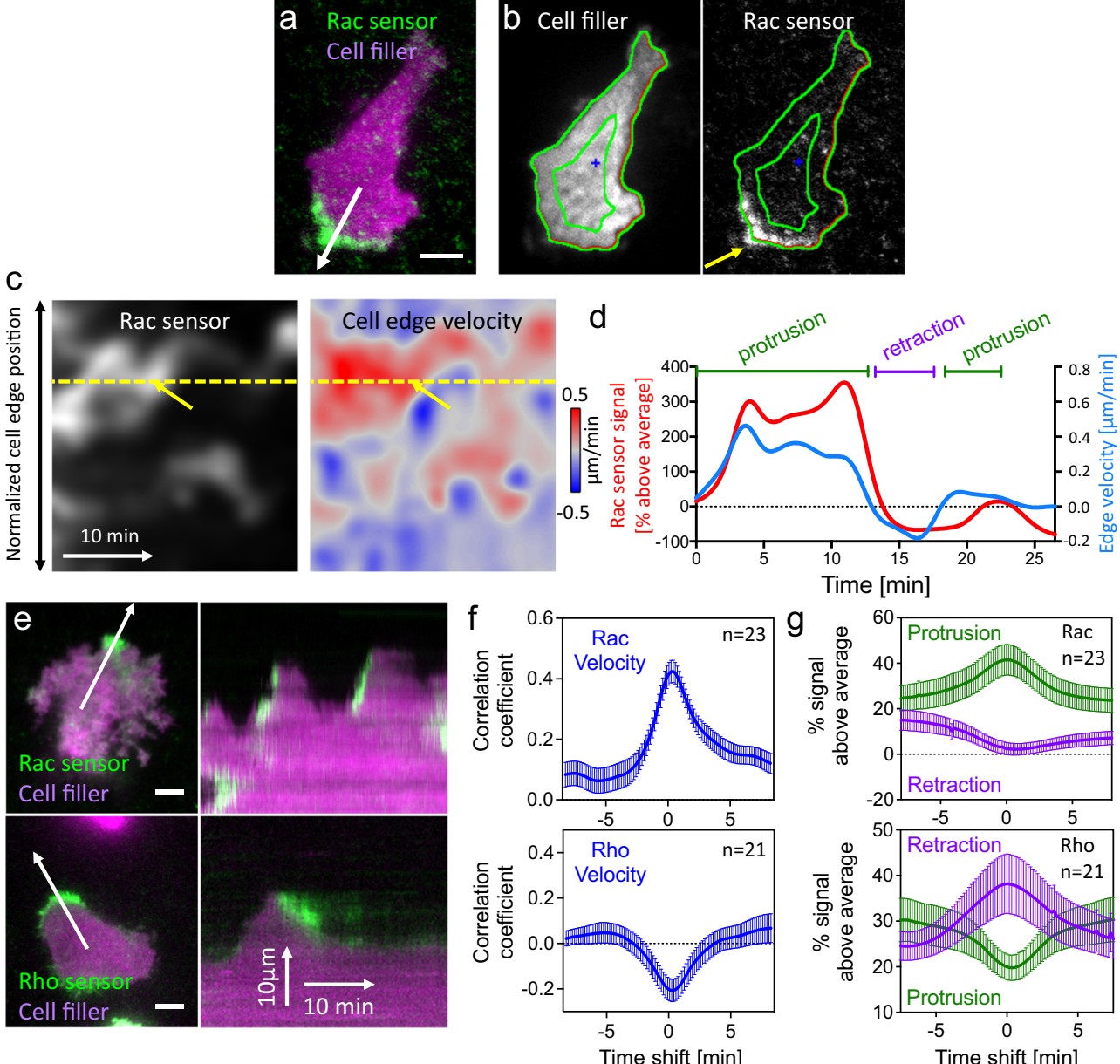

**Fig. 4 | Sequential Rac and Rho activation is tightly coupled to cell protrusion-retraction cycles in space and time. a** Representative A431 cell expressing a cytosolic cell volume marker (mCitrine) and an improved Rac activity sensor (mCherry-3xp67$^{phox}$GBD; top; see also Supplementary Movie 3; $n = 23$ cells from 3 independent experiments). The white arrow marks the direction of a local cell protrusion. **b** Automated tracing of the cell border using a modified version of the ADAPT plugin[43]. **c** Maps generated by the ADAPT plugin that represent the spatio-temporal dynamics of the Rac sensor signal between the green lines in (**b**) (left) and of the cell edge velocity (right). The yellow arrows in (**c**) point to the local protrusion that occurs at the position of the cell area marked by the yellow arrow in (**b**). Red areas in the velocity map correspond to local cell protrusions, blue areas to local cell retractions. **d** Plot of Rac sensor signals and cell edge velocity corresponding to the yellow dotted line in (**c**). **e** Representative TIRF images (left) of

A431 cells that generate spontaneous protrusion-retraction cycles and express the Rac or Rho GTPase activity sensors and the cell volume marker (see also Supplementary Movies 4 and 5). White arrows represent the protrusion direction. Kymographs (right) correspond to white arrows in TIRF images. **f** Crosscorrelation between Rac sensor signal and cell edge velocity plotted against the time shift between these measurements. **g** Enrichment of Rac and Rho sensor signals in protrusions (>0.075 μm/min) and retractions (<−0.075 μm/min). Values are normalized to average control sensor enrichment measurements. $n = 3$ independent experiments with >21 cells per condition. Error bars represent standard error of the mean. Measurements corresponding to individual cells are shown in Supplementary Fig. 5c, d. Scale bars: 10 μm; 0.26 μm/pixel. PH Pleckstrin homology domain, DH Dbl homology domain, Source data are provided as a Source Data file.

In the original implementation, regions are considered for measurement that extend equally both inside and outside the cell border. We changed these analysis areas to only include signals approximately 3 μm towards the inside of the cell border to increase the signal measurement sensitivity (the modified plugin is available via Github, see methods for details). We then used this modified plugin to extract spatio-temporal fluorescence signal and cell edge velocity maps (Fig. 4c) and temporal signal-cell edge velocity cross-correlation functions (Fig. 4f). The measured correlation between Rac and cell edge velocity supports the observation that Rac activity increases during cell protrusion (see also Supplementary Movies 3 and 4). Conversely, the anti-correlation measured for Rho supports the

observation that Rho activity increases during cell retraction (negative cell edge velocity, see also Supplementary Movie 5). Furthermore, the maxima and minima of these functions are only minimally shifted in time, showing that the correlation between sensor signal and cell shape changes does not have a significant delay.

The correlation functions contain limited information about the relation between the sensor signals and cell shape changes. First, they cannot be used to distinguish between a signal increase during protrusion and a signal decrease during retraction. Both of these events would result in a positive cross-correlation value. However, cell protrusion and cell retraction are very distinct cellular processes that involve distinct sets of regulators. Therefore, measurements that mix these processes such as the signal/cell edge velocity correlation shown in Fig. 4f, blur the information specific to these distinct processes. Second, the correlation functions do not provide a measure for the strength of the signal, and how much it is enriched in a particular area of the cell. To overcome these limitations, we developed our own, extended analysis approach, in which we used the signal and velocity maps (Fig. 4c) to measure, how much the fluorescence signal is enriched within local cell protrusions or retractions relative to the average signal of the whole cell (Fig. 4d, the analysis script is available via Github, see methods for details). Analogous to temporal cross-correlation functions, we introduced time shifts between the protrusion-retraction events and signal measurements, to obtain temporal signal enrichment functions that show how much the fluorescence signal is enriched or depleted relative to the time periods of protrusion and retraction (Fig. 4g). The enrichment of the activity signal is calculated in percent relative to the average signal of the whole cell attachment area, which is an easily interpretable measure for local sensor signal strength.

Applying this analysis to the cell protrusion regulator Rac1 yielded a very clear representation of the observed dynamics (Fig. 4g; top panel), i.e., that Rac1 activity is highly enriched during cell protrusion and slightly depleted during retraction. Conversely, activity of the cell retraction regulator Rho was highly enriched during cell retraction and depleted during protrusion (Fig. 4g; bottom panel, see also Supplementary Movie 5).

Together, these observations show that the dynamic protrusion-retraction cycles that are observed in A431 cells are tightly coupled to corresponding activations of Rac and Rho. Based on these observations, we hypothesized that the Rac-dependent activation of Rho, which we identified in this study, might play a role in the tight coupling of signals in dynamic protrusion-retraction cycles.

**Molecular mechanism of sequential Rac/Rho activity dynamics**
To investigate this hypothesis, we sought to identify potential mediators of Rac-dependent Rho activation. A previous biochemical study suggested that Rho activators of the Lbc GEF family might act as effectors of active Rac (Fig. 5a) and thereby might mediate this activity crosstalk[44]. In this hypothetical mechanism, active Rac1 would recruit an Lbc GEF to the plasma membrane, thereby concentrating the GEF at this site to stimulate local Rho activity. To narrow down the set of potential candidates, we investigated the plasma membrane association dynamics of all Lbc-type GEFs in relation to cell protrusion and retraction dynamics. In this focused screen, we found that a subset of the Lbc GEFs: Arhgef11 (PDZ-RhoGEF) and Arhgef12 (LARG), were highly enriched at the edge of the cell during protrusion and retraction (Fig. 5b, c, see also Supplementary Movies 6 and 7).

We next used our analysis approach to quantify the enrichment dynamics of Arhgef11 and Arhgef12 at the cell edge relative to cell protrusion and retraction events (Fig. 5d). Similar enrichments of Arhgef11/12 during cell protrusion were measured using a plasma membrane marker as control (Supplementary Fig. 5b). These analyses showed that both GEFs were maximally enriched shortly after cell

protrusion. By combining measurements of Rac and Rho activity as well as GEF plasma membrane recruitment, we were able to derive a temporal sequence of events (Fig. 5e), in which enrichment of active Rac is maximal during cell protrusion with minimal delay (~0 s), followed by the Lbc-type GEFs Arhgef11 (~160 s) and Arhgef12 (~210 s) and lastly active Rho (>470 s).

Similar observations were made using temporal cross-correlation functions (Supplementary Fig. 5a). This places maximal Arhgef11 and Arhgef12 enrichment between maximal Rac and Rho activation, which further supports a role for these Lbc-type GEFs in mediating Rac/Rho GTPase activity crosstalk in the coordination of cell protrusion-retraction cycles. To directly investigate the causal relationship between Rac, Arhgef11/12 and Rho, we again used the light-controlled PA-Rac1 construct[20] and combined optogenetic, rapid Rac1 activation with measurements of plasma membrane recruitment of wild-type or mutant Arhgef11/12 constructs (Fig. 6a). With this approach, we were able to directly show that both Arhgef11 and Arhgef12 are rapidly recruited to the plasma membrane after optogenetic Rac1 activation (Fig. 6b, c). Taken together, these results suggest that Arhgef11 and Arhgef12 are able to link cell protrusion and retraction by mediating activity crosstalk between Rac and Rho.

Several domains in Arhgef11/12 could be involved in the Rac1 activity-induced plasma membrane recruitment (Fig. 6a). An interaction between active Rac1 and the PH domains of Arhgef11/12 as suggested above, would represent a particularly simple mechanism. In addition, Arhgef11 can also bind F-actin[45], which could indirectly be involved in its recruitment to the cell cortex near the plasma membrane via Rac1-stimulated actin polymerization. Less direct mechanisms could also be envisioned, for example via Galpha12/13, which can interact with the ARHGEF11/12 RGS domain[46,47], or by heterodimerization, which was previously reported for Arhgef11/12[48]. To investigate the underlying mechanism more directly, we first compared the plasma membrane recruitment of wild-type Arhgef11/12 with constructs that include point mutations that were previously shown to interfere with the Rac1 interaction of the related Lbc-type GEF p190RhoGEF[44]. As shown in Fig. 6b, c, we find that an intact PH domain of Arhgef12 is required for efficient plasma membrane recruitment and thus suggests that this domain plays an important role in the Rac1-stimulated recruitment of this GEF.

In contrast, corresponding mutations in the PH domain of Arhgef11 did not reduce plasma membrane targeting, suggesting that another part of this molecule is responsible (Fig. 6b, c). Interestingly, in contrast to Rho, which is activated at the plasma membrane homogenously following optogenetic Rac1 activation (Fig. 3g), the recruitment of Arhgef11 and Arhgef12 differed between the central and peripheral cell attachment areas. This was particularly evident for Arhgef11, which was initially recruited much stronger in the central cell attachment area, and subsequently particularly enriched in peripheral cell attachment areas (Fig. 6d). We reasoned that these more complex dynamics might be associated with the ability of Arhgef11 to interact with F-actin, which is expected to increase after Rac1 activation. Indeed, deletion of the Arhgef11 F-actin binding site reduced both the initial response in the cell center and the delayed response in the cell periphery (Fig. 6e, f). These observations suggest that Arfgef12 is recruited to the plasma membrane via its PH domain, while Arfgef11 is recruited to the cell cortex near the plasma membrane by its interaction with increased amounts of F-actin downstream of active Rac1. As plasma membrane recruitment was not completely inhibited by these mutations, heterodimerization between Arhgef11/12 or binding to other potential interaction partners, such as Galpha12/13, might play an additional role.

To investigate, if Arhgef11 or Arhgef12 are required for Rac/Rho activity crosstalk, we used RNA interference to reduce their expression level. As shown by western blot analysis, we were able to efficiently

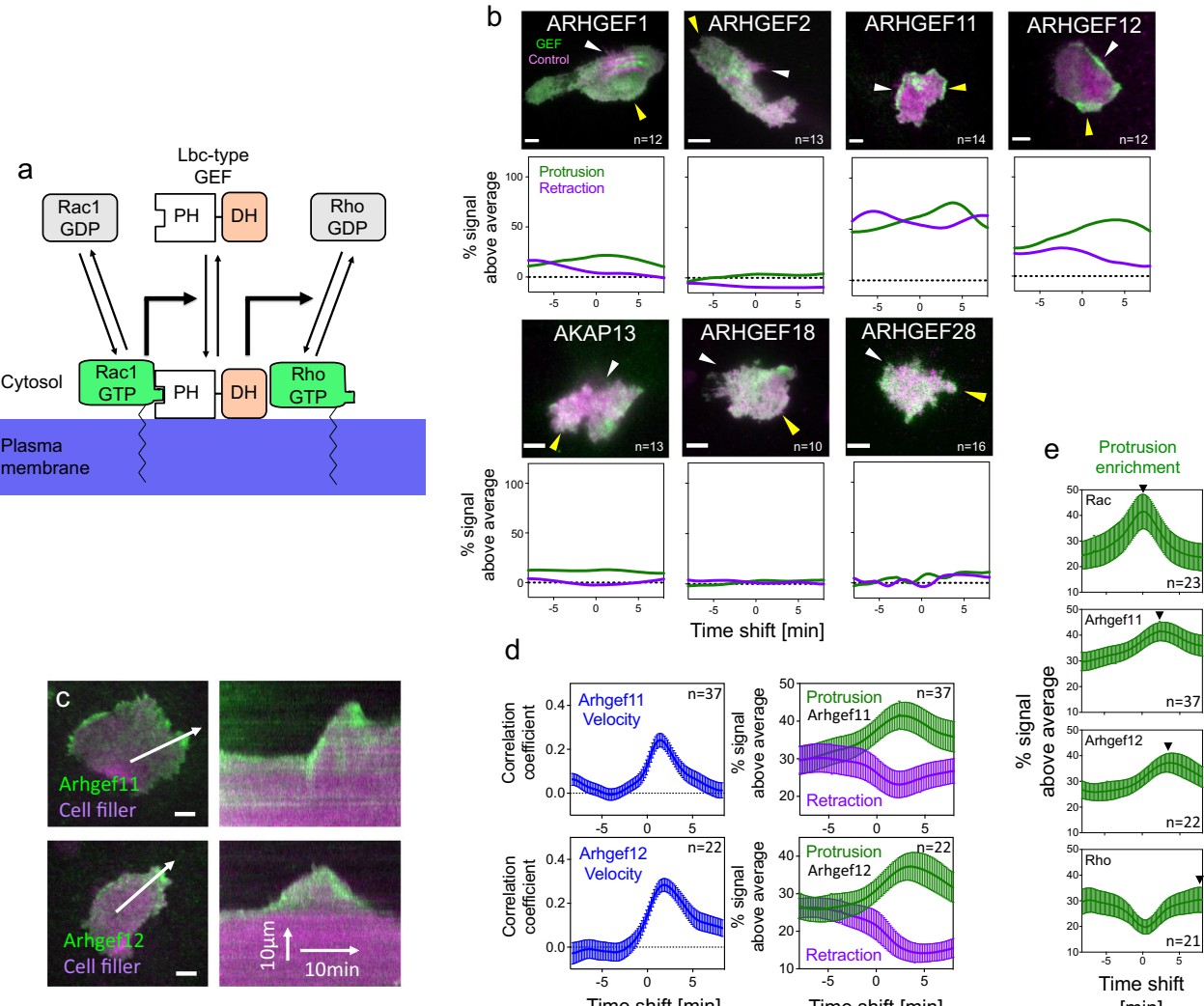

**Fig. 5 | Identification of Arhgef11 and Arhgef12 as Rac effectors in local cell protrusion-retraction cycles. a** Schematic representation of a hypothetical mechanism, by which Lbc-type GEFs could mediate Rac1/Rho activity crosstalk. **b** TIRF microscopy images (top panels) and protrusion-retraction enrichment functions (bottom panels) for representative cells that express Lbc-type GEFs (CMV-GEF, green) and a cytosolic cell volume marker that acts as a control construct (delCMV-mCitrine, magenta; *n* > 10 cells from 3 independent experiments). White and yellow arrows point to local cell retractions and protrusions, respectively. **c** Representative TIRF images (left) of A431 cells that generate spontaneous protrusion-retraction cycles and express Arhgef11 and Arhgef12 fused to mCherry and the cytosolic cell volume marker (mCitrine, see also Supplementary Movies 6 and 7). White arrows represent the protrusion direction. Kymographs (right) correspond to white arrows in TIRF images. **d** Crosscorrelation (left) between

Arhgef11/Arhgef12 signals and cell edge velocity plotted against the time shift between these measurements, and enrichment (right) of Arhgef11/Arhgef12 signals in protrusions and retractions. Arhgef11/Arhgef12 enrichment values are normalized to average control construct enrichment measurements. *n* = 3 independent experiments with >22 cells per condition **e** Direct comparison of signal enrichment of active Rac, Arhgef11, Arhgef12 and active Rho relative to the time period of cell protrusion. Black arrows indicate the time point of maximal sensor or GEF enrichment. The measurements shown in this panel are identical to measurements shown in Fig. 4g and Fig. 5d. Error bars represent standard error of the mean. Measurements corresponding to individual cells for panels (**d**) and (**e**) are shown in Supplementary Fig. 5c, d. PDZ PSD-95 Dlg ZO-1 domain, RGS Regulator of G protein signaling domain, FAB F-actin binding domain, Source data are provided as a Source Data file.

knock down Arhgef11 protein (up to 83 ± 15%) via this strategy and indeed find that this knockdown strongly decreased Rac1-stimulated Rho activation (Fig. 6g, h, Supplementary Fig. 6a–c). Knockdown of Arhgef12 was less efficient (up to 75 ± 11%), but the effect was similar (6g, h, Supplementary Fig. 6a–c).

**Arhgef11/Arhgef12 in cell protrusion/retraction dynamics**

To test, if these GEFs indeed play a role in cell protrusion-retraction cycles and associated cellular processes, we investigated how altering their expression level might affect these dynamic processes. Conceptually, if Arhgef11/12 indeed can mediate Rac/Rho crosstalk, increasing their expression level should increase this crosstalk and thereby shorten the time period that links cell protrusion and

retraction. Indeed, ectopic expression of Arhgef11 or Arhgef12 significantly decreased the duration of protrusion-retraction cycles (Fig. 7a, b).

Conversely, decreasing the expression level of these GEFs would be expected to weaken the link between protrusion and retraction and thus slow down protrusion-retraction cycles. To test this, we again used RNA interference to reduce the expression of theses GEFs. In these experiments we indeed find that the knockdown of Arhgef11 or Ahrgef12 slowed down protrusion-retraction cycles (Fig. 7c, d and Supplementary Fig. 6d). Taken together, these findings suggest that Arhgef11/12 mediate Rac/Rho activity crosstalk, and that this crosstalk plays a role in the dynamic interplay between cell protrusion and cell retraction dynamics.

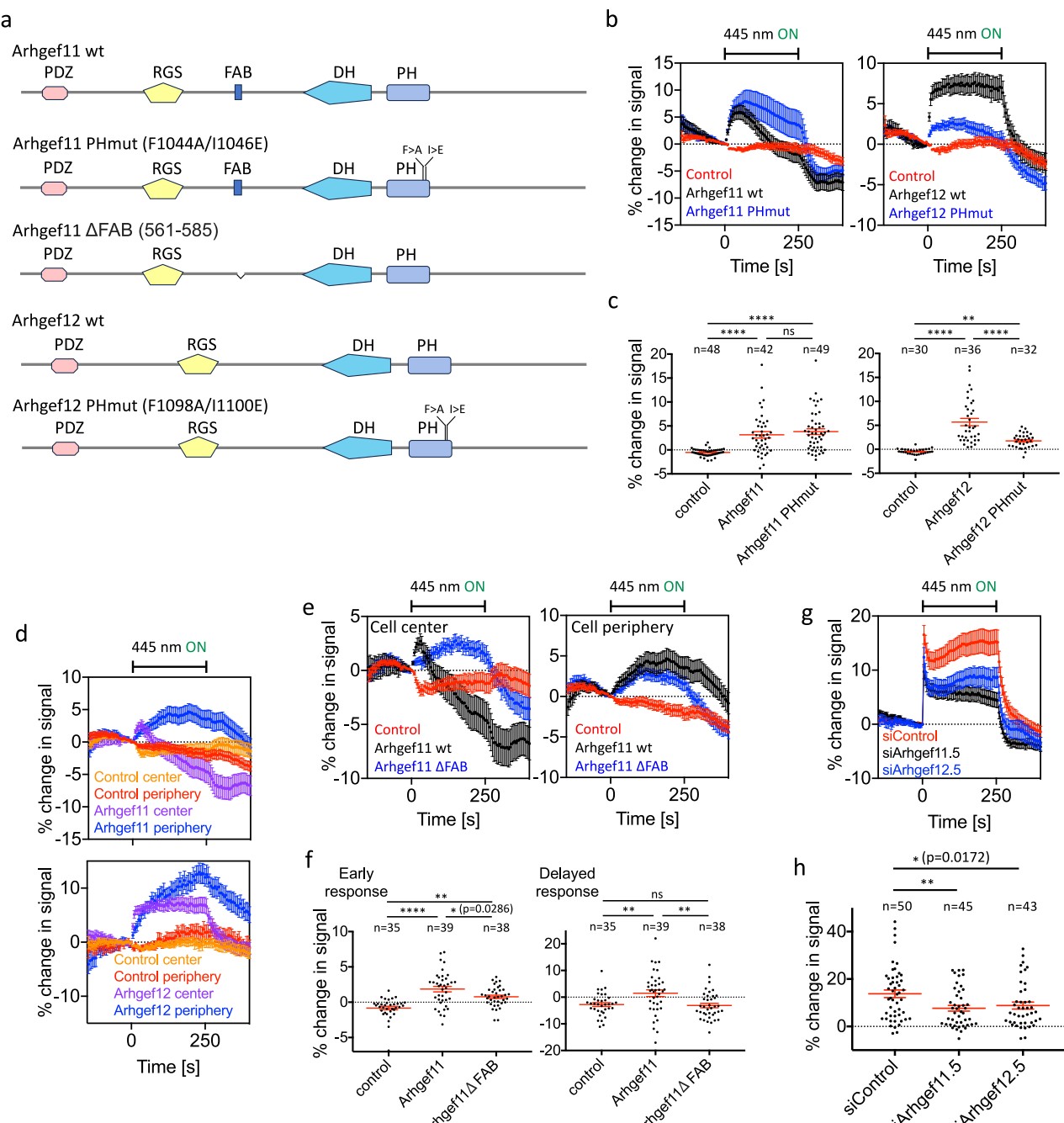

**Fig. 6 | Characterization of Rac-dependent Arhgef11 and Arhgef12 plasma membrane recruitment. a** Schematic representation of the Arhgef11 constructs that were used investigate the mechanism of Rac-stimulated plasma membrane recruitment. **b**–**f** Measurement of full length or mutant Lbc-type GEF plasma membrane recruitment during Rac activation in A431 cells that co-express PA-Rac1 (*n* = 3 independent experiments). **b**, **d**, **e** Measurement of recruitment kinetics. **c**, **f** Quantification of recruitment in the 25 s time frame during photoactivation (early response) or 1 min after photoactivation (late response). **d** Measurement of Arhgef11/12 plasma membrane recruitment in peripheral vs central cell attachment areas. **g**, **h** Quantification of average Rho activity sensor kinetics before, during and after Rac1 activation in A431 cells that co-express the Rho activity sensor, PA-Rac1 and control or Arhgef11/12 targeting siRNA oligonucleotides. **g** Measurement of average Rho activity sensor kinetics, corresponding to data shown in (**h**). **h** Quantification of the Rho activity response in the 25 s time frame during photoactivation. ****$P < 0.0001$; **$P < 0.01$; *$P < 0.05$; One-way ANOVA with Tukey's (**c**, **f**) or Holm-Sidak's (**h**) post test. Error bars represent standard error of the mean. Scale bars: 10 μm; 0.26 μm/pixel. All statistical tests were two-sided. Source data are provided as a Source Data file.

## Role of Arhgef11/Arhgef12 in cell migration

Dynamic cell protrusion-retraction cycles are associated with the exploratory cell migration mode typically observed in A431 cells. We therefore investigated, if the frequent cell shape changes that are mediated by Arhgef11/12 contribute to efficient cell migration. Indeed, tracking of individual cells showed that loss of these GEFs significantly decreased migration distance (Fig. 7e, f and Supplementary Fig. 6e)

and therefore reduced their ability to explore their environment. Interestingly, the total displacement was less affected by Arhgef11/Arhgef12 knockdown (Fig. 7e, g) and even increased with one of the Arhgef11 siRNAs (Supplementary Fig. 6f), suggesting that the reduced exploratory migration was associated with an increase in migration directionality. Indeed, the ratio of displacement to distance, a typical measure for directionality of migration trajectories, was significantly

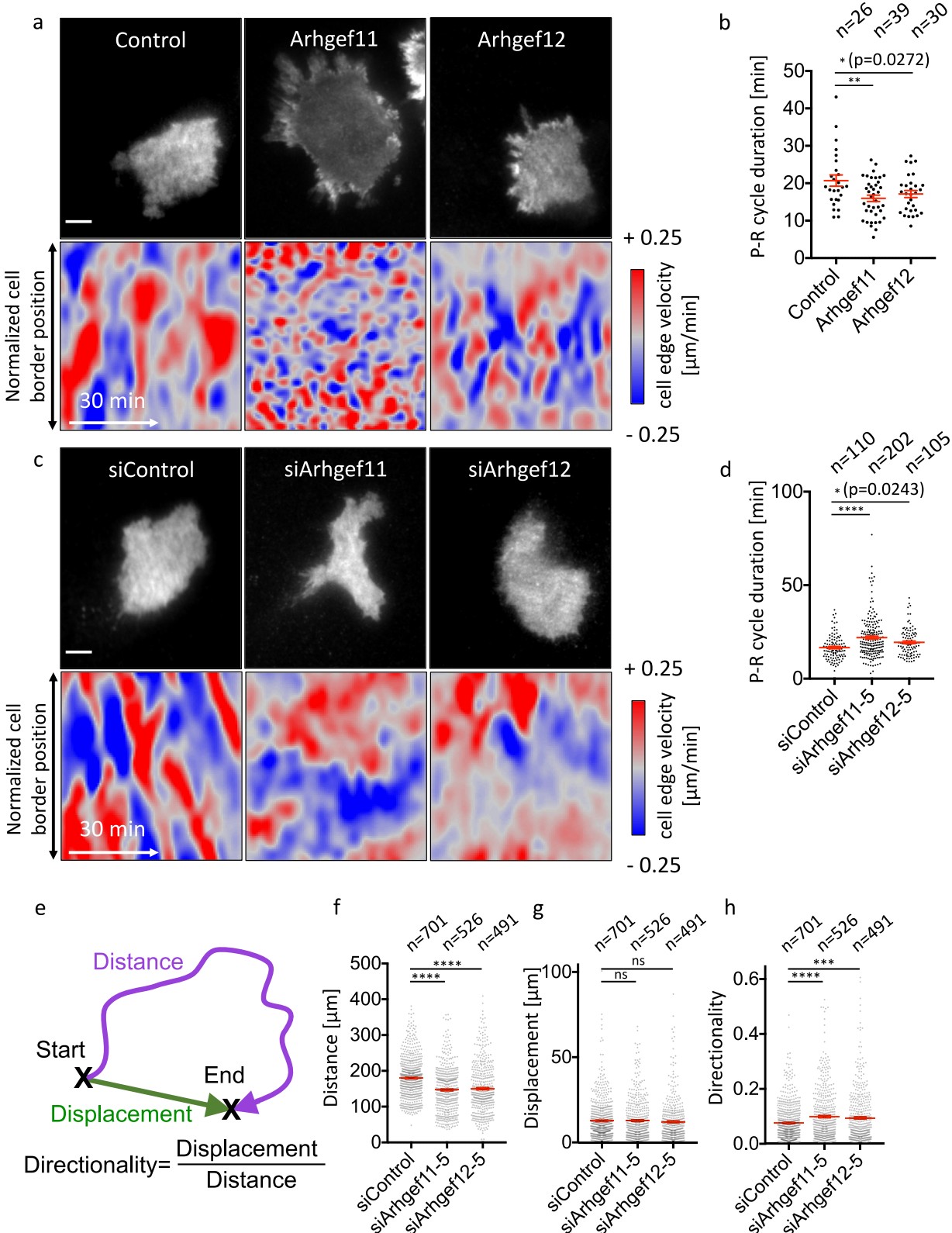

increased after Arhgef11 or Arhgef12 knockdown (Fig. 7e, h and Supplementary Fig. 6g).

## Discussion

In this study we investigated the signal crosstalk between the major Rho GTPases Rac, Rho and Cdc42. These investigations surprisingly revealed that rapid activation of the cell protrusion regulator Rac1

leads to an increase in the activity of the cell contraction regulator Rho. This crosstalk was observed in several cell lines, however, the detailed kinetics of the Rho activity response differed significantly (Fig. 3d, e and Supplementary Fig. 3a, b). In the neuroblast-derived N2a cells, Rho activity increased over a prolonged time period, whereas keratinocyte-derived A431 cells showed a more transient response that quickly adapted during the Rac1 perturbation. As the perturbation is based on

**Fig. 7 | Arhgef11 and Arhgef12 mediate Rac-dependent Rho activation and the spatio-temporal coordination of local cell protrusion-retraction cycles.** **a**–**d** Quantification of protrusion and retraction dynamics in A431 cells with increased (**a**, **b**) or decreased (**c**, **d**) Arhgef11 and Arhgef12 expression levels. **a**, **c** TIRF images (top panels) and cell edge velocity maps (bottom panels) of representative cells that express CMV-mCherry-Arhgef11/Arhgef12 and delCMV-mCitrine (**a**), or Arhgef11/Ahrgef12 targeting siRNA and delCMV-mCherry (**c**). **b**, **d** Quantification of protrusion-retraction (P-R) cycle duration based on cell edge velocity measurements corresponding to panels (**a**) and (**c**), respectively (**a**, **b** $n = 3$ independent experiments with >26 cells per condition, **c**, **d** $n = 3$ independent experiments with >105 cells per condition). Differences in the average values obtained for the two control conditions are presumably due to the experimental protocols, which differ significantly between (**b**) and (**d**) (single vs. dual transfection; see "Methods" for details). **e** Schematic representation of distance (magenta) and displacement (green) for typical spontaneous exploratory cell migration. The distance corresponds to the length of the cell migration trajectory which leads to the indicated displacement between the start and end locations. The directionality is defined as the ratio between these length measurements. **f**–**h** Quantification of distance (**f**), displacement (**g**) and directionality (**h**) of A431 cell trajectories over a 4 h time period in control and Arhgef11/Arhgef12 depleted cells ($n = 3$ independent experiments with >491 cells per condition). (*$P < 0.05$; **$P < 0.01$; ***$P < 0.001$; ****$P < 0.0001$; One-way ANOVA with Holm-Sidak's post test). Images were recorded at a frame rate of 1.5/min (**a**–**d**) or 1/min (**f**–**h**). Error bars represent standard error of the mean. Scale bars: 10 μm; 0.26 μm/pixel. All statistical tests were two-sided. Source data are provided as a Source Data file.

a dominant positive Rac1 mutant, which cannot be inhibited itself, this adaptation has to occur downstream in the crosstalk mechanism. One possible mechanism is self-inhibition of Rho via a delayed negative feedback, for example by Myosin[17,49]. Cell type specific differences in this feedback could account for the observed differences in the response dynamics. Similar feedback mechanisms could also lead to adaptation in other crosstalk mechanisms, for example in the observed transient inhibition of Rac by RhoA (Fig. 2d, e).

To characterize the function of the Rac/Rho crosstalk in cells, we focused on highly dynamic cell protrusion-retraction cycles that are typically observed during spontaneous, mesenchymal cell migration. Our detailed analysis revealed that the activity of Rac and Rho was tightly coupled with phases of cell protrusion and retraction, respectively. This is in contrast to previous studies using FRET sensors, which suggested that Rho is most active during cell protrusion, followed by active Rac[8]. This difference could be explained by considering the mechanisms, by which distinct sensor constructs detect activity[50]. FRET-base sensors detect the ratio of GEFs vs GAPs, while translocation-based sensors detect the amount of active endogenous GTPase at the plasma membrane[50]. The parameter that is most relevant for activity crosstalk between Rho GTPases is the amount of active endogenous GTPase, which is what we measure with our translocation-based sensors. Furthermore, our findings are in line with the established biochemical functions of Rac and Rho[36–39]. The close coupling between Rac and Rho with opposing dynamic cell shape changes suggests, that activity crosstalk between these GTPases might play a role in coordinating cell protrusion and cell retraction. Conceptually, coupling of Rac and Rho activity could be mediated by several distinct mechanisms. Our direct investigation shows that Rac activates Rho, and that conversely, Rho inhibits Rac. Such a link would represent a negative feedback mechanism, which is distinct from the often-cited idea that Rac and Rho mutually inhibit each other[4,51].

The idea of mutual inhibition between Rac and Rho appears quite attractive, as it can be used to explain cell polarization of migrating cells, in which a Rac-dependent protrusive front region and a Rho-dependent contractile back region mutually exclude each other. Indeed, a mechanism based on mutual inhibition alone is expected to stably segregate protrusive front and contractile back signals as it is observed in persistent directional cell migration of neutrophils[5]. However, additional mechanisms would be required to enable dynamic cycling of protrusive and contractile signals as it is observed during exploratory migration of mesenchymal cells[11]. The oscillatory or excitable system dynamics observed for protrusive front and contractile back signals could play a role in the formation of such cycles of protrusion and retraction. However, to coordinate these cycles, front and back signals have to be coupled with each other. The asymmetric hierarchy, which is implied by our crosstalk analysis, i.e. activation of Rho by Rac and inhibition of Rac by Rho, is expected to enable the temporal cycles of protrusions followed by retractions, which are typically observed in such cells. A signal network that is based on mutual inhibition between Rac and Rho instead would lock a local cell area either in a protrusive or retractile state and would require an additional signal to dynamically switch local cell shape changes.

While the concept of mutual inhibition between Rac and Rho is frequently cited[4,52–55], several earlier studies suggested a potential activating role of Rac on Rho. Most prominently, initial studies of cellular Rho GTPase function showed that injection of an active Rac mutant induced formation of stress fibers that were dependent on Rho activity[56]. To investigate this mechanism further, we focused on the recent finding that members of the Lbc family of Rho GEFs might mediate Rac-dependent Rho activation[44]. While biochemical data in that study suggested Arhgef28 (p190RhoGEF) as a particularly strong candidate for mediating this crosstalk, we did not detect any enrichment of Arhgef28 at cell protrusions or retractions in A431 cells. However, a focused screen of all Lbc family GEFs showed that Arhgef11/12 are highly enriched at the peripheral edge of these cells, both during cell protrusion and during cell retraction, making these two molecules interesting candidates to mediate Rac/Rho crosstalk. By combining optogenetic Rac activation with TIRF microscopy, we indeed found that both Arhgef11 and Arhgef12 are Rac effectors.

Interestingly, the recruitment of Arhgef11/12 by active Rac1 reached a maximum at a later time compared to the downstream activation of Rho. This difference might be due to the enzymatic action of the GEFs, which could very efficiently activate many Rho molecules once they are present at the membrane. In addition, positive feedback amplification of Rho activity[17], could further accelerate the response. Furthermore, subsequent negative feedback, for example via Rho self-inhibition[17] can subsequently reduce Rho activity. The combination of positive and negative feedback could therefore explain the rapid formation of the transient Rho pulse that we frequently observe during Rac1 photoactivation, and that peaks earlier compared to Arhgef11/12 (Fig. 3c–e).

The observation of Rac1-dependent plasma membrane recruitment of Arhgef11/12 suggests that these Rho GEFs might be able to coordinate dynamic cell shape changes associated with Rac and Rho activity. Increasing or decreasing the expression level of either Arhgef11 or Arhgef12 revealed that these molecules indeed play a role in the coupling between dynamic cell protrusions and retractions.

Together with previous work, we propose a mechanism, in which Arhgef11/12 couple the signal modules that control cell protrusion and cell retraction (Fig. 8). We showed that active Rac can recruit Arhgef11/12 (Fig. 6b, c), which are well known to activate Rho[46,47]. Increased Rho activity can be further amplified via positive feedback[17], and can inhibit Rac activity (Fig. 2d, e). A role for Rho as a Rac inhibitor is further supported by the observation that Rho activity is minimal right at the onset of cell protrusion (Fig. 5e). Thus, Rho itself must be inhibited to enable another protrusion-retraction cycle. This Rho self-inhibition could be mediated by previously proposed negative feedback loops[17].

Finally, our study revealed how Arhgef11/12 are linked to exploratory cell migration. By coupling Rac/Rho activity, Arhgef11/12 are expected to mediate cell retraction after protrusion and thus facilitate spontaneous changes in migration direction. Indeed, we find

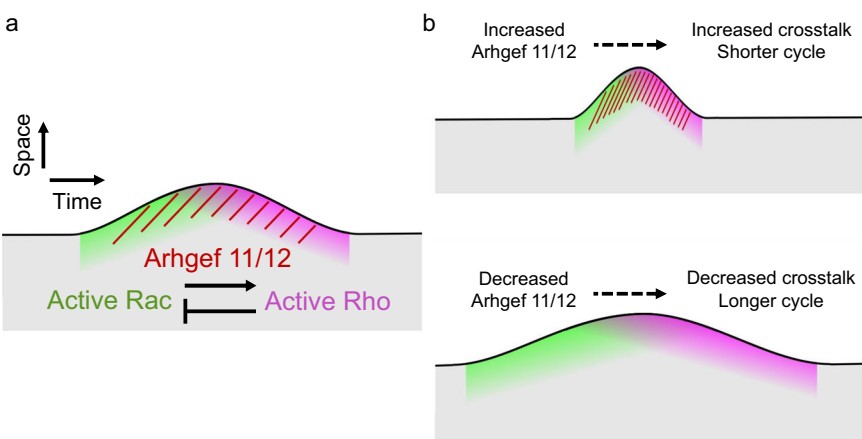

**Fig. 8 | Proposed mechanism for the generation of protrusion-retraction cycles. a** Schematic for spatio-temporal events that couple cell protrusion and retraction and the signal molecules that mediate this coupling. **b** Effect of increasing or decreasing Arhgef11/12 levels on local Rac/Rho crosstalk and cell morphodynamics.

that knockdown of Arhgef11/12 increased directionality during cell migration (Fig. 7h; Supplementary Fig. 6g). Thus, our study reveals a mechanism, how crosstalk between the major cell morphogenesis regulators Rac and Rho links local cell protrusion and retraction dynamics to enable effective exploratory cell migration.

## Methods

### Cell culture and reagents

Neuro-2a cells (ACC 148, DSMZ, Braunschweig) were cultured in MEM Eagle (10% FBS, 100 U/ml Penicillin + 100 µg/ml Streptomycin, 2 mM L-Glutamine, 1 mM Sodium Pyruvate, PAN Biotech). A431 cells (CRL-1555, ATCC) were cultured in DMEM medium (10% FBS and 2 mM L-Gluta-mine, PAN Biotech). HeLa (kind gift from Prof. Hemmo Meyer, University Duisburg-Essen), NIH3T3 (CRL-1658, ATCC), and U2OS cells (HTB-96, ATCC) were cultured in DMEM medium (10%FBS and 50 U/ml Penicillin + 50 µg/ml Streptomycin, 2 mM L-Glutamine, PAN Biotech). All cells were maintained using standard culture techniques at 37 °C and 5% $CO_2$. For live-cell imaging, Neuro-2a, HeLa, and NIH3T3 cells were plated onto LabTek glass surface slide (Thermo Fischer Scientific) or glass-bottom dishes (MatTek) before transfection of plasmid DNA. Pre-treatments of glass surfaces were as follows: uncoated for Neuro-2a and Hela cells, poly-L-lysine for NIH3T3 cells and 10 µg/ml Collagen-I for U2OS cells. A431 cells were plated on LabTek glass surfaces that were coated with 10 µg/ml Fibronectin for 45 min. The SLF'-TMP dimerizer and TMP competitor were synthesized as described previously[19]. To quantify GEF/sensor enrichment at the cell periphery, cells were plated on 35 mm culture dishes, transfected with plasmid DNA, and then replated on fibronectin-coated MatTek glass-bottom dishes (10 µg/ml fibronectin for 16 h at 4 °C).

### Plasmid constructs and siRNA

EGFP-2x-FKBP'-Rac1Q61LΔCAAX and TagBFP-2xeDHFR-CAAX for targeting active human Rac1 to the plasma membrane via chemically-induced dimerization were described previously[19]. Here, optimized constructs for targeting of Rac1, Cdc42 and RhoA were prepared. First, EGFP in EGFP-2x-FKBP'-Rac1Q61LΔCAAX was replaced by mTurquoise2 to facilitate simultaneous measurement of the four fluorophores TagBFP, mTurquoise2, mCitrine and mCherry. In addition, a nuclear export sequence was introduced between mTurqouise2 and FKBP' to prevent nuclear accumulation, which results in delayed plasma membrane recruitment. In brief, mTurquoise2-2xFKBP'-Rac1Q61LΔCAAX was generated by ligating the larger fragment from EGFP-2xFKBP'-Rac1Q61LΔCAAX[19] with the smaller fragment from pmTurquoise2-N1 (Addgene Plasmid #60561), after digestion with BsrGI and NheI.

In a second step, this resulting plasmid and a PCR fragment amplified from pmTurquoise2-NES[57] (Addgene Plasmid# 36206) with primers 5'-TCAGTTGCTAGCCTCAAGCTTCGAATTCTG-3' and 5'-AGAGTCAGCTC-GAGATATCTTGTACGAGTCCAG-3', were digested using NheI/XhoI. The larger fragment of the plasmid was ligated to the PCR product to yield the final perturbation construct mTurquoise2-NES-2xFKBP'-Rac1Q61LΔ-CAAX. The analogous human Cdc42 and RhoA perturbation constructs were generated as derivatives from this Rac1 construct by ligation to PCR fragments amplified from pcDNA3-EGFP-Cdc42Q61L or pcDNA3-EGFP-RhoAQ63L (kind gifts from Gary Bokoch, The Scripps Research Institute) using 5'-CTGTACTCTAGATCCATGCAGACAATTAAGTG-3'/5'-TCGAGTCAATTGAGTTAGGACCTGCGGCTCTTC-3' and 5'-GGAATTC-TAGATCCATGGCTGCCATCCGGAAG-3'/5'-CGAGTCAATTGAGTTAGGAA CCAGATTTTTTC-3', respectively, after digestion of both fragments using MfeI/XbaI. The plasmid coding for mCherry fused to β-actin, driven by the Ubiquitin-C promotor (mCherry-actin-Ub) was generated in two steps: 1. mCherry-C1-Ub was generated from pUB-GFP (Addgene plasmid #11155) by inserting mCherry amplified from mCherry-C1 (Clontech) using 5'-CGGCATTAATGATCTGGCCTCCGCGCCGGGT-3' and 5'-GGCCGCTAGCCGACCTGCAGCCCAAGCTTCGTC-3', after digestion of both fragments with AseI/NheI, 2. mCherry-actin-Ub was generated from mCherry-C1-Ub by inserting β-actin amplified from mRFP1-actin[58] using 5'-ATATGAATTCCGCCCCCGCGAGCACAGA-3' and 5'-AT ATGGATCCTCAGTGTACAGGTAAGCCCTGGC-3', after digestion of both fragments with EcoRI/BamHI. The Rac, Cdc42 and Rho activity sensor constructs driven by the low expressing delCMV promotor[59], delCMV-mCherry-p67phox-GBD, delCMV-WASP-GBD and delCMV-mCherry-RhotekinGBD were described previously[17]. The control construct delCMV-mCitrine was generated by ligating the larger fragment from pmCitrine-N1 (Addgene Plasmid #54594) with the smaller fragment from the delCMV-mCitrine-RBD sensor[17], after digestion with AseI and BsrGI. delCMV-mCherry was generated by ligating the larger fragment from pmCherry-N1 (Clontech) with the smaller fragment from delCMV-mCherry-actin[60], after digestion with AseI and BsrGI. mCerulean-PA-Rac1Q61L was a kind gift from Klaus Hahn (University of North Carolina). mCherry-NMHCIIA[61] was obtained from Addgene (Plasmid #35687). The majority of transfections were performed using Lipofectamine™2000 (Thermo Fisher Scientific). Transfections of A431 cells were performed using Lipofectamine™3000 (Thermo Fisher Scientific). For initial experiments in Neuro-2a cells (Fig. 1), XtremeGene 9 (Roche Diagnostics) was used.

The improved Rac1 activity sensor (delCMV-mCherry-3X-p67phox-GBD) was generated from the previously established delCMV-mCherry-p67phox-GBD construct[17]. The sensor was generated in two

steps via PCR based Gibson assembly. For the Gibson reaction, a mix was prepared consisting of 5% PEG-8000 (Promega), 100 mM Tris-HCl (pH 7.5), 10 mM MgCl2, 10 mM DTT, 0.2 mM dATP, 0.2 mM dTTP, 0.2 mM dCTP, 0.2 mM dGTP, 1 mM NAD (New England Biolabs), 2.0 U T5 exonuclease (New England Biolabs), 12.5 U Phusion DNA polymerase (New England Biolabs), 2000U Taq DNA ligase (New England Biolabs). First, the p67phox-GBD insert was amplified using 5′- GGACTCAGATCTCGAGCTCACATGTCCCTGGTGGAGGCCA-3′/5′- ACCAGGGACATGGAATTCGATCCACTTCCAGAACCCGTGCGCCTTGCCTAGGTAATC-3′ and inserted into the parent plasmid after linearization using HindIII to generate delCMV-mCherry-2x-p67phox-GBD. In the second step, one p67phox-GBD repeat was amplified from delCMV-mCherry-2x-p67phox-GBD using 5′-ACGGGTTCTGGAAGTGGATCGGTTCTCATGTCCCTGGTGGAGGC-3′/5′-GGCCTCCACCAGGGACATGGAATTCGATCCACTTCCAGAACCCGTGCGCCTTGCCTAGGTAATC-3′ and inserted into the EcoRI-cut delCMV-mCherry-2x-p67phox-GBD plasmid to generate delCMV-mCherry-3x-p67phox-GBD. The Rho activity sensor (delCMV-mCherry-2X-RhotekinGBD) was generated using the previously established delCMV-mCherry-RhotekinGBD construct[17]. The sensor was generated using an analogous PCR based Gibson assembly as described above. First 1x-RhotekinGBD was amplified using 5′-TACAAGTCCGGACTCAGATCTCGAGAAGCTTCGAATTCCCTGG-3′/5′-AGGGAATTCGAAGCTTGAGCGAGTCCGGAGCCTGTCTTCTCCAGCAC-3′. The amplified fragment was then inserted into the at XhoI-cut parent plasmid.

CMV promoter-driven plasmids encoding human Lbc GEFs (CMV-mCherry-Arhgef1, CMV-mCherry-Arhgef11, CMV-mCherry-Arhgef12, CMV-mCherry-AKAP13, CMV-mCherry-Arhgef18, and CMV-mCherry-Arhgef28) were gifts from Oliver Rocks[62]. CMV-mCherry-Arhgef2 was generated using PCR based Gibson assembly by replacing Arhgef1 in the CMV-mCherry-Arhgef1 plasmid with human Arhgef2. First, Arhgef2 (also known as GEF-H1) was amplified from the previously established EGFP-GEF-H1[63] plasmid using 5′-AGTCCGGACTCAGATCTCGAGGGCGCGCCATGTCTCGGATCGAATCCC -3′/5′- GATCCGGTGGATCCTTAGTTAATTAAGCTCTCGGAGGCTACAGC-3′. The amplified fragment was then inserted into CMV-mCherry-Arhgef1, after removing the Arhgef1 insert flanked by AscI and PacI sites.

A delCMV promoter-driven Arhgef1 encoding plasmid (delCMV-mCherry-Arhgef1) was generated using Gibson assembly by amplifying the protein coding sequence from the CMV-mCherry-Arhgef1 plasmid using 5′-CTCCACCGGCGGCATGGACGAGCTGTACAAGTCCGGACTCAGATCTC 3′/5′ TCTAGAGTCGCGGCCGCTTTACTTTTAGTTAATTAAAGTGCAGCCAG 3′. The amplified fragment was then inserted into a BsrGI cut delCMV-mCherry plasmid. delCMV-mCherry-Arhgef1 was then used to generate delCMV-mCherry-Arhgef11 by replacing Arhgef1 with Arhgef11 from CMV-mCherry-Arhgef11 at AscI/PacI sites. delCMV-mCherry-Arhgef12 was generated using Gibson assembly by amplifying Arhgef12 from CMV-mCherry-Arhgef12 using 5′-GCTGTACAAGTCCGGACTCAGATCTCGAGGGCGCGCCAGTGG-CACACAGTCTACTATC-3′/ 5′-CCGCTTTACTTTTAGTTAATTAAACTTTTATCTGAGTGCTTGTC-3′. The amplified fragment was then inserted into delCMV-mCherry-Arhgef1, after removing the Arhgef1 insert flanked by BglII and PacI.

The control construct delCMV-mCitrine-CAAX was generated by ligating an adapter based on two oligonucleotides (5′-GTACAGTGGATCCGGCGGTTCCGGTAAGATGAGCAAAGATGGTAAAAAGAAGAAAAAGAAGTCAAAGACAAAGTGTGTAATTATGTAAGC-3′/5′-GCCGCTTACATAATTACACACTTTGTCTTTGACTTCTTTTTCTTCTTTTTACCATCTTTGCTCATCTTACCGGAACCGCCGGATCCACT-3′) with BsrGI/NotI-cut delCMV-mCitrine[17].

Arhgef11 PHmut (F1044A/I1046E) was generated using Gibson assembly by amplifying two protein coding DNA fragments from CMV-mCherry-Arhgef11 using two sets of primers (5′-ATAAGACCTTGGACCTCCACGTGCTGCTGCTGGAGGACCTC-3′/5′-ATTCGATAGCGAAGGCCCGTTTATCTGTGG-3′ and 5′-ACGGGCCTTCGCTATC GAATGCACCTCCAAGCTGGGC-3′/5′-TAGGAACTTACCTGGTTAATTAATGGTCCTGGTGACGCGGC-3′). These two fragments were then inserted into CMV-mCherry-Arhgef11 cut by PacI/PmlI. Arhgef12 PHmut (F1098A/I1100E) was generated similarly using Gibson assembly by amplifying two protein coding DNA fragments from CMV-mCherry-Arhgef12 using two sets of primers (5′- CAGCGAGTATCCAGAGAAGGAATTCTGTCACCCTCAGAGCTAC-3′/5′- ATTCGACGGCTAAAGCTTTGTTATCTGTTGCC-3′ and 5′- CAAAGCTTTAGCCGTCGAATCCATGTCAGACAATGGC-3′/5′- TAGGAACTTACCTGGTTAATTAATCAACTTTTATCTGAGTGCTTG-3′). These two fragments were then inserted into CMV-mCherry-Arhgef11 cut by PacI/EcoRI-HF.

Arhgef11 ΔFAB(561-585) lacking the F-actin binding sequence from amino acids 561 to 585[45] was generated using Gibson assembly by amplifying protein coding DNA fragments from CMV-mcherry-Arhgef11 using 5′-CCCTTTTTTTTCCCCAGGGGCGCGCCATGAGTGTAAGGTTACCCC-3′/5′-AGAGTCGTTGGACTTCCACAGGGGACAAG-3′ and 5′-TGTGGAAGTCCAACGACTCTCGACCGGAAG-3′/5′-AGGTCCTCCAGCAGCAGCACGTGGAGGTCCAAGGTCTTATCCTTG-3′. These two fragments were then inserted into CMV-mCherry-Arhgef11 cut with PmlI/AscI.

For knockdown experiments, ON-Target plus siRNAs (Dharmacon™) were used (siControl: #2 5′- UGGUUUACAUGUUGUGUGA-3′, siArhgef11: #5 5′-GCAAGUGGCUGCACAGUUC-3′, #7 5′-UCUAUGAGCUGGUUGCAUU -3′, siArhgef12: #5 5′-GAUCAAAUCUCGUCAGAAA-3′, #6 5′-GAAAUGAGACCUCUGUUAU-3′). Briefly, cells were transfected with 30 nM of the siRNAs using Lipofectamine™ RNAiMAX (Invitrogen). Cells were incubated with the siRNAs for 48 h before splitting and reseeding onto glass-bottom dishes. For subsequent microscopy experiments, the reseeded siRNA-treated cells were transfected a second time using the Lipofectamine 3000 reagent for expression of the reporter plasmid constructs. Experimental analysis of knockdown phenotypes was performed a total of 96 h post siRNA treatment and quantification of knockdown efficiency was performed at that same timepoint via Western blot analysis.

All newly generated plasmid constructs will be available from the Addgene repository after publication.

## Western blot analysis

Cells were washed one time with ice cold PBS, then lysed with ice cold 1× cell lysis buffer (9803, CST) for 5 min on ice. The cell lysate was then centrifuged at 17000 × $g$ for 10 min at 4 °C to remove insoluble material. A Bradford assay was used to measure the protein concentration in the supernatant. 5× Laemmli sample buffer was used to prepare protein samples, which were boiled at 95 °C for 5 min before being separated using SDS-PAGE (4561086, Biorad).

Wet blot transfer was used to transfer proteins to a PVDF membrane (MERCK). Intercept blocking buffer (927-60001, LI-COR) was used to block the membrane for 60 min at room temperature.

Blots were incubated for 24 h with primary antibodies at 4 °C while shaking (Arhgef 12 antibody GTX87286, Lot no. 822105710 at 1:1000; Arhgef11 antibody sc-166740, Lot no. I21 10 at 1:50; GAPDH antibody CST-2118, Lot. No. 14 as loading control at 1:1000). All antibodies were diluted in intercept blocking buffer. The membranes were then washed with TBS-T buffer and stained with secondary antibodies (IRDye 680RD Goat anti-Mouse IgG, 926-68070, Lot no. D10901-15; IRDye 800CW Goat anti-Rabbit IgG, 926-32211, Lot no. D10831-15, LI-COR at 1:10,000) for 60 min at room temperature. After final washing steps, the blots were measured using the Odyssey® CLx imaging system (LI-COR).

## Microscopy

TIRF microscopy was performed on an Olympus IX-81 microscope, equipped with a TIRF-MITICO motorized TIRF illumination combiner, an Apo TIRF 60 × 1.45 NA oil immersion objective and a ZDC autofocus device. For the majority of experiments that employed spectral

emission ranges in blue (TagBFP), cyan (mTurquoise2/mCerulean), yellow (mCitrine) and red (mCherry), a dichroic mirror (ZT405-440/514/561) was used in combination with an emission filter set (HC 435/40, HC 472/30, HC 542/27 and HC 629/53), the 514 nm OBIS diode laser (150 mW) (Coherent, Inc., Santa Clara, USA) or the 514 nm line of a 400 mW Argon ion laser (model # 543-A-A03, Melles Griot, Bensheim, Germany), and the Cell R diode lasers (Olympus) with wavelength 405 nm (50 mW), 445 nm (50 mW) and 561 nm (100 mW), as well as wide-field illumination via the MT20 light source (Olympus) or the Spectra X light engine (Lumencor). To avoid any potential effect from Förster resonant energy transfer (FRET), we always only excited one fluorophore at the time. For detection, this was combined with an EMCCD camera (C9100-13; Hamamatsu, Herrsching am Ammersee, Germany) at medium gain without binning. The microscope was equipped with a temperature-controlled incubation chamber. Time-lapse live-cell microscopy experiments were carried out with indicated frame rates at 37 °C in $CO_2$-independent HEPES-stabilized imaging medium (Pan Biotech) supplemented with 10% FBS.

Tracking of single cell migration was performed using an Olympus IX-81 microscope with a UPlanSApo 10× objective. Wide-field images were acquired using a 651 nm LED lamp with Spectra X light engine (Lumencor). The microscope was equipped with a temperature-controlled incubation chamber. Time-lapse live-cell microscopy experiments were carried out at 37 °C in $CO_2$-independent HEPES-stabilized imaging medium (Pan Biotech) supplemented with 10% FBS.

To measure the effect of rapid Rac1 perturbation on cell volume, Neuro-2a cells were co-transfected with a volume marker (delCMV-mCherry) and the small molecule based perturbation system (mCitrine-2 × eDHFR-tKRas and mTurquoise2-NES-2 × FKBP'-Rac1Q61LΔCAAX or the empty control perturbation construct mTurquoise2-NES-2 × FKBP'). Image stacks were acquired using the Zeiss LSM510 microscope equipped with a 63× C-Apochromat Objective (NA 1.2). The 561 nm line from a DPSS 561-10 diode laser and the 514 and 455 nm laser lines of an Argon laser were used to excite mCherry, mCitrine and mTurquoise fluorophores, respectively. The change in cell volume was measured using the 3D watershed function of the 3d ImageJ Suite[64] via Fiji.

## Perturbation via chemically induced dimerization

Chemically-induced dimerization was performed essentially as described before[19]. Briefly, synthesis and purification of the dimerizer SLF'-TMP and of the competitor TMP followed established protocols[19]. Neuro-2a cells were transfected with TagBFP-2xeDHFR-CAAX and mTurquoise2-NES-2xFKBP' fused to the Q61LΔCAAX mutant of Rac1, Cdc42 or RhoA. To investigate morphological changes, cells were co-transfected with mCherry-actin-Ub, to investigate Rho GTPase crosstalk, cells were co-transfected with the Rac, Cdc42 or Rho sensor constructs delCMV-mCherry-p67$^{phox}$-GBD, delCMV-WASP-GBD or delCMV-mCherry-RhotekinGBD, respectively. In all experiments, the delCMV-mCitrine control sensor was co-expressed. Chemically-induced dimerization was initiated by addition of 10 µl SLF'-TMP dimerizer and stopped by addition of 10 µl TMP competitor.

## Optogenetic perturbation

Cells were transfected with photo-activatable Rac1 (mCerulean-PA-Rac1) and light-based activation of mCerulean-PA-Rac1 was performed by TIRF illumination using the 445 nm Cell R diode laser. To prevent saturated PA-Rac1 activation, 1000x-10000x neutral-density filters were added into the 445 nm TIRF illumination light path. The built-in neutral-density filter wheel was used to fine-tune light intensity and was typically set to ~30%. Within photoactivation time intervals, 445 nm TIRF illumination was constantly on, except for the exposure times during image acquisition. To minimize background activation of mCerulean-PA-Rac1, illumination intensity, duration and frequency

were kept as low as possible and fluorescence measurements were always performed using the 514 nm and 561 nm excitation lines, including the excitation of EGFP fluorophores. Detection of TagBFP, mTurquoise2 or mCerulean was always performed after the experiment. Compared to perturbations via chemically-induced dimerization, which require additional time for compound uptake by cells, optogenetic Rac1 activation was very fast. We therefore focused our investigations on the immediate crosstalk response with this method.

## Measurements of subcellular morphodynamics

To investigate the local enrichment of Lbc-type GEFs in cell protrusions and retractions, A431 cells were transfected with plasmids coding for mCherry fused Lbc GEFs driven by the CMV promotor. To improve the signal-to background ratio, the automated quantification of Arhgef11 and Arhgef12 signals at the cell periphery was performed by transfecting mCherry-GEF constructs driven by the delCMV promoter. To study the local enrichment of the GTPase activity in cell protrusions and retractions, cells were transfected with the enhanced Rho or Rac sensor. To study the effect of overexpression of Arhgef11 and Arhgef12 on cell morphodynamics, cells were transfected with plasmids coding for mCherry fused GEFs driven under CMV promoter. In all experiments described above, delCMV-mCitrine was co-transfected to identify the cell edge independent of the signals of interest. To study the effect of siRNA mediated knockdown on cell morphodynamics, delCMV-mCherry was used to identify the cell borders.

## Measurements of cell migration

The effect of siRNA knockdown on cell migration was investigated by staining nuclei in living A431 cells using the SPY-650-DNA dye (Spirochrome). 96 h post siRNA treatment, cells were seeded on 10 µg/ml fibronectin-coated LabTek dishes at a density of ~143 cells/mm². 6 h post seeding, cells were incubated with SPY-650 (1:1000) in imaging medium for 1.5 h. Cells were then washed once with fresh imaging media and rested for 1 h before onset of imaging. Fluorescence measurements were performed using the 651 nm LED lamp of the SpectraX light engine. Images were collected with a frame rate of 1/min.

## Image and video analysis

All image analysis was performed using ImageJ (http://imagej.nih.gov/ij/) and all figures were prepared using Photoshop CS4. Kymograph analysis was performed using the ImageJ built-in multi kymograph plugin. Cell tracking was performed using the Trackmate plugin. Statistical analysis, curve fitting (mono-exponential decay) and generation of plots was performed using Prism (GraphPad). All statistical tests were two-sided.

## Analysis of Rho GTPase activity crosstalk via chemically-induced dimerization

The fluorescence intensity of Rho activity sensor $I_{Rho}$ was measured via TIRF microscopy in central cell regions that were completely adherent during the entire observation period. During the time course of perturbation via chemically-induced dimerization, cell shrinkage or expansion occurred that lead to Rho activity independent changes in fluorescence intensity, presumably due to changes in cell volume and associated changes in protein concentration. To correct for these intensity changes, we co-expressed a control sensor (delCMV-mCitrine) and measured the fluorescence intensity of the control sensor $I_{Control}$ in identical cell regions as the intensity of the Rho activity sensor. These raw intensity measurements were normalized by subtracting the background signals outside cell areas $I_{Rho,BG}$ and $I_{Control,BG}$ and dividing by the initial, background-corrected intensity values $I_{Rho,0} - I_{Rho,0,BG}$ and $I_{Control,0} - I_{Control,0,BG}$ before the perturbation. The corrected activity sensor measurements $A_{Rho,corr}$ were obtained by subtracting the normalized control sensor measurements from the

normalized Rho GTPase activity sensor measurements:

$$A_{Rho,corr} = \frac{I_{Rho} - I_{Rho,BG}}{I_{Rho,0} - I_{Rho,0,BG}} - \frac{I_{Control} - I_{Control,BG}}{I_{Control,0} - I_{Control,0,BG}} \quad (1)$$

### Analysis of Rac/Rho GTPase activity crosstalk via optogenetic perturbations

Optogenetic perturbation of Rac1-induced only minor changes in the control sensor signal during the short time periods of the crosstalk measurements (see Fig. 3 and Supplementary Fig. 3a–f). Therefore, subtraction of control sensor measurements was not necessary in these experiments and activity sensor measurements $A_{Rho}$ were calculated via the following simplified equation:

$$A_{Rho} = \frac{I_{Rho} - I_{Rho,BG}}{I_{Rho,0} - I_{Rho,0,BG}} \quad (2)$$

In these experiments, delCMV-mCitrine or delCMV-mCherry were used as control sensors, which were either co-expressed with the Rho sensor (N2A, NIH3T3, Hela or U2OS cells), or expressed in a separate cell population (A431 cells).

The central and peripheral cell attachment areas were obtained using the distance map function of ImageJ. The peripheral part included the outer 5 pixels, which corresponds to a region of 1.3 μm around the cell circumference. The central part included all other pixels of the cell attachment area.

### Analysis of cell morphodynamics and local fluorescence signals at the cell edge

Analysis of local cell edge velocity and associated local fluorescence signals were performed using a modified version of the ADAPT plugin[43] in combination with custom ImageJ analysis scripts. In brief, the published ADAPT plugin performs analysis of local signal intensities via a region that extends from the cell edge both inwards, in the direction of the cell center and outwards, away from the cell. Thereby, signals that are localized near the cell edge are decreased by averaging with the local background intensity and therefore do not permit quantitative estimations of relative, local signal enrichment. To enable such estimations, the ADAPT code was modified to generate a region that only extends from the cell edge inwards. The signal and cell velocity maps and signal/velocity cross-correlation functions were directly extracted from the standard output of the modified ADAPT script. To calculate the local signal enrichment, spatio-temporal regions in the cell velocity maps that correspond to cell protrusion (>0.075 μm/min) or retraction (<−0.075 μm/min) were selected via the threshold function and transferred to the fluorescence signal maps. The average intensity within these regions was measured and the enrichment was calculated as the ratio of this intensity divided by the average intensity in the whole cell (including the cell edge and central cell regions) over the full time period. To generate the local signal enrichment functions, the spatio-temporal regions described above were shifted along the temporal x-axis of the fluorescent signal map, to extract the time-shifted local signal enrichment. To calculate the duration of protrusion-retraction cycles, the cell periphery was divided into 100 positions. This was achieved by rescaling the cell velocity maps in the position axis to a size of 100 pixels. In each horizontal line of the rescaled cell velocity maps, time periods were identified as protrusions or retractions. Protrusion-retraction cycle duration was defined as the time period starting from the onset of a protrusion and the following onset of a retraction. For experiments to investigate the effect of increasing Arhgef11/12 levels via ectopic expression or decreasing Arhgef11/12 levels via RNAi, the threshold to identify protrusions or retractions was the same as for the signal enrichment analysis (>0.075μm/min for protrusions and <−0.075 μm/min for retractions).

### Analysis of cell migration velocity and directionality

Quantification of velocity and directionality was performed using the TrackMate plugin[65] in ImageJ. SPY-650 stained nuclei were selected and segmented using an intensity threshold and size filter. Cells that leave the field of view during tracking were not included in the analysis. Tracks generated by TrackMate were used to calculate velocity and directionality measurements using the chemotaxis plugin (Ibidi GmbH, Martinsried, Germany) in ImageJ.

### Reporting summary

Further information on research design is available in the Nature Portfolio Reporting Summary linked to this article.

## Data availability

The data generated in this study which are shown in scatter plots and line plots in main and supplementary figures are provided in the Source Data file. The raw data generated in this study for measurement of the Rac and Rho activity dynamics in migrating cells shown in Fig. 4a–d and Supplementary Fig. 4a–d are provided in a Github repository along with associated analysis code (https://github.com/agdehmelt/protrusion_retraction_enrichment_analysis). All other datasets generated during and/or analyzed during this study are available from the corresponding author on request. Source data are provided with this paper.

## Code availability

Code for the adjusted Adapt plugin and the protrusion/retraction signal enrichment analysis presented in this study is available via Github repositories: https://github.com/agdehmelt/protrusion_retraction_enrichment_analysis[66]. https://github.com/agdehmelt/adapt_edge_analysis_modification[67].

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

## Acknowledgements

We thank Sven Müller (MPI Dortmund) for expert microscopy support and Philippe Bastiaens (MPI Dortmund) for departmental support and helpful discussions. We also would like to thank Stefanie Gossen for generating the delCMV-mCitrine-CAAX construct, Lukas Grebe for support in plasmid generation, and Ricarda Lüttig for support in cell manipulations. This work was supported by MERCUR grant No. Pr-2012-0022 to P.N. and L.D., FORSYS partner initiative (BMBF grant No. 0315258) to L.D., the Deutsche Forschungsgesellschaft DFG project grants 823/3-1 and 823/9-1 to L.D., DFG Heisenberg Programme grants 823/4-1, 823/6-1, 823/8-1 to L.D., DFG Principal Investigator grant DE 823/10-1 to L.D., a DAAD Ph.D. fellowship to A.S., the European Research Council, ERC (ChemBioAP) to Y.W.W., the Knut and Alice Wallenberg Foundation to Y.W.W., the Göran Gustafsson Foundation for Research in Natural Sciences and Medicine to Y.W.W. and Vetenskapsrådet (Nr. 2018-04585) to Y.W.W. DFG CRC1430 subproject A08 Project-ID 424228829 to P.N.

## Author contributions

L.D., P.N., S.N., A.C, A.S., and T-T.D. designed the research. S.N. performed, analyzed, and optimized the majority of experiments to investigate protrusion-retraction dynamics in A431 cells and the role of GEFs in mediating Rac/Rho crosstalk. A.C. developed, characterized, and optimized the activity perturbation and the activity measurements and performed initial Rac1-Rho crosstalk analyses. A.S. performed, analyzed, and optimized experiments to investigate the mechanism how active Rac recruits Arhgef11/12 to the plasma membrane or cell cortex. T-T.D. performed, analyzed, and optimized the majority of combined Rho activity perturbation-response measurements. S.N. contributed to characterization of activity perturbation method development. S.N. developed improved Rac and Rho activity sensors based on tandem GBDs. J.K. performed initial combined perturbation-response analysis and contributed to combined perturbation-response analysis. D.S. contributed to activity sensor development. Y.W.W. and X.X. contributed to activity perturbation development and synthesized the chemical dimerizer. L.D. and P.N. jointly supervised experiments. L.D. and P.N. wrote the majority of the manuscript. All authors contributed to discussions and manuscript preparation.

## Funding

## Competing interests

The authors declare no competing interests.
