## [Peer Review File · Nature Communications]

Rho GTPase activity crosstalk mediated by Arhgef11 and Arhgef12 coordinates cell protrusion-retraction cyclesREVIEWER COMMENTS

Reviewer #1 (Remarks to the Author):

In this interesting study, the authors investigate the crosstalk between the GTPases, Rac and Rho. In the first part of the paper, the authors used and optimized two previously developed assays to study the crosstalk of Rho GTPases in N2A cells. Using these assays, they discover that artificial Rac activation causes Rho activation. In the second part of the paper, they use A431 cells to understand the mechanism of the regulation of Rho activation by Rac. Using a cross-correlation analysis approach, they find that Rac GTPase is activated before Rho GTPase during a protrusion event. They identify the RhoGEFs Arhgef11 and Arhgef12 as novel regulators of the Rac-mediated activation of Rho. They modulate the expression level of Arhgef11 and Arhgef12 by overexpression and knockdown and identify 2 phenotypes to explain the effect of Arhgef11 and Arhgef12 on the Rac-mediated activation of Rho.

Major points. The manuscript should in my view be published if the concerns below can be addressed:

- 1) The authors need to show data and discuss if there are spatial or temporal constraints for the mechanism of Rac to Rho activation they describe. For example, is the mechanism they describe only happening at the front of a migrating cell or also at the back of a migrating cell? Can the authors comment on how their assay is affected by measuring signal at the cell edge vs. the center of the cell?
- 2) In figure 2 and 3, the authors show that Rho is activated upon Rac stimulation. They need to show whether knockdown of Arhgef11 and Arhgef12 change the activation of Rho. I recommend that the authors use the dimerization-based or optogenetics-based approach and show in more detail the effect of Arhgef11 and Arhgef12 knockdowns on changes in the strength and duration of Rho activation.
- 3) In figure 6, the authors show some effect of Arhgef11 and Arhgef12 knockdowns on protrusion-retraction cycle duration. However, the effect of siRNA control on reducing protrusion-retraction cycle duration is as strong as the effect of the Arhgef11 or Arhgef12 overexpression. The authors need to include data using a WT control and discuss why siControl may have such an effect. Similarly, the authors should include a WT control for the data presented in figure 6f-g.
- 4) To test if Arhgef11 and Arhgef12 work synergistically, the authors could do a double knockdown of Arhgef11 and Arhgef12.

Minor points

Figure 1 is presented in a confusing way as it introduces 2 fluorescent markers but then in 1B does not show how the signal of the reporter looks like but instead shows an Actin reporter. Please add the actual image data of the reporter in this figure and make clear which reporters are used.

In figure 2C, they should use the same scale for the control sensor corrected response signal in all plots.

In figure 2, did the authors correct for possible FRET interaction of their BFP and YFP reporter?

In figure 3, the authors use a different duration for the perturbation, 250s (4.17 min) instead of 20 min in figure 2. Can they comment on why they chose a different duration?

In figure 3, they only show the Rho activation data for the A431 cell line. However, all experiments so far have used the N2A cells. Can the authors include the Rho activity dynamics over time for these cells as well?

In figure 3C, there are 4 different scenarios of Rho activity, upon Rac activation, presented. The 3 lower panels (in sum 71% of the cells) on the right all have a downward trend of Rho activity from the pre-stimulus baseline signal after perturbation with Rac activity. Can the authors present the data using the same scale and comment on this effect?

In figure 5e, the authors show that there is a temporal sequence of events with Rac peaking first, then Arhgef11 and Arhgef12, then Rho. However, Rho activity is also decreasing while Rac and Arhgef11 and Arhgef12 are already increasing. Can the authors comment on the decline of Rho activity during that period? Also, the section in which Rho is reaching its peak is cut off from the figure. It seems that Rho activity is still increasing.

What is the reasoning for the time scale of Rac-dependent activation of Rho? Can the authors give discuss why the Rac-dependent activation of Rho may last for +20 min while the inhibitory effect of Rho on Rac is only apparent at 5 min after Rho activation?

The authors should use standard deviation for indicating the variability of their data, in addition/instead of standard error of the mean throughout the paper.

Reviewer #2 (Remarks to the Author):

Review of “Crosstalk between Rac and Rho GTPase activity mediated by Arhgef11 and Arhgef12 coordinates cell protrusion-retraction cycles” by Nanda et al.

The cell cortex is patterned by Rho GTPases during a variety of fundamental biological processes such as cell locomotion, cell division, and cell repair. In many situations, crosstalk between the three most-studied Rho GTPases—Rho, Rac, and Cdc42—is thought to contribute to such cortical patterns.

In the current study, the authors address cortical patterning during the protrusion-retraction cycle of cell locomotion. To track Rho GTPase dynamics, the authors used probes based on diffusion trapping, in which soluble, Fluorescent protein (FP) GTPase activity reporters are recruited from a cytoplasmic pool to the plasma membrane at sites of GTPase activation. This is not a new approach (see also below) as it was developed nearly 20 years ago for studies of cell repair and cytokinesis. However, for reasons that are not at all clear, studies of cell locomotion have usually relied on FRET-based reporters which, strictly speaking, do not measure endogenous GTPase activity.

The authors first show that experimental Rac activation by two different approaches induces Rho activation in several different cell lines. This is both a surprising and an exciting finding. It’s surprising in that it has generally been thought that Rac and Rho are mutually antagonistic during cell locomotion. It’s exciting because it suggests a simple (on paper, anyway) link between protrusion and retraction: Rac activation first leads to protrusion but then, due to (presumably) a delayed activation of Rho, retraction will naturally ensue.

The authors then image Rac and Rho activity during natural cell locomotion. Consistent with the above findings, they see that cortical pulses of Rac activity immediately precede protrusion while similar pulses

of Rho activity immediately precede retraction. Finally, the authors identify two GEFs, Arhgef11 and Arhgef12, as potential links between Rac and Rho.

This is important work that will be of considerable interest to students of Rho GTPase signaling and cell locomotion. For the former, the results challenge a treasured dogma—that Rac antagonizes Rho and vice versa. Similarly, the successful deployment of the simple GBD-based probes to study cell motility may change the impression that such probes are inherently inferior to FRET-based probes. For the latter, the model developed by the authors provides an elegant, plausible explanation for how protrusion can be coupled to retraction.

I have no major concerns with the study, but there are several areas that could do with some revision or further explanation. These are presented below:

A. While it is clear that ectopic activation of Rac induces Rho activation in most cell types, it was not so clear whether there is a consistent delay between the two and for how long the delay lasts. Presumably, the effect cannot be instantaneous, but I was simply unable to determine this from the data presented.

B. The authors have done a nice job revealing Rho pulses following Rac activation; one cannot help but wonder if depletion of Arhgef11 or Arhgef12 (or both) reduces the amplitude of these pulses. However, I recognize this is an easy experiment to request on paper but may be quite challenging in practice, as periodic behaviors such as the authors are studying can vary considerably under control conditions.

C. Based on the abstract and the introduction, I had the impression that the authors developed improved Rho activity sensors in this paper. However, upon closer reading of the manuscript, I learned otherwise: The authors state that “...we used TIRF microscopy-based sensor constructs that we developed previously (Graessl et al., 2017)”. I took this to mean that the probes are somehow specifically useful for TIRF and that they were developed by the authors. However, upon checking the Graessl paper, it turns out that the Rho and Cdc42 probes were simply FPs fused to Rhotekin rGBD and NWASP GBD, respectively. The FP-rGBD approach was developed for live cell imaging in 2005 as described in PMID: 15684032. The FP-wGBD probe approach was developed for live imaging in 2003 by two different labs more or less simultaneously (PMIDs: 12874226 and 12872130). These probes do indeed work with TIRF but they also work with a variety of other imaging modalities and as far as I can tell they were not modified for TIRF in either Graessl et al. or in the current study.

Later, the authors state that they increased the sensitivity of the probes by generating “...constructs that contain tandem GTPase binding domains that could benefit from binding multiple GTPase molecules at the same time. This alternative sensor design might lead to increased interactions with active GTPases due to increased avidity” and then “Using these improved sensors...”. The implications here are that the authors show that these sensors are more sensitive and, given the context, that this is something that hasn't been tried before. However, nowhere in the paper is there any kind of comparison, quantitative or otherwise, of the original sensors (based on a single GBD) to the “improved” sensors (based on two GBDs). With respect to novelty, the authors use mCherry-2X-rGBD as an improved sensor for active Rho. This exact construct was used in 2016 (PMID: 27226483). With respect to demonstration of

improvement, while not in the current manuscript, a systematic study from 2021 showed introduction of two or more GBDs into Rho biosensor probes improves their sensitivity (PMID: 34357388).

This is not to say that the choice of probes doesn't matter because it does, as the authors make clear in the Discussion. Nor is it to say that this paper is somehow less noteworthy because the probe technology is not new. Indeed, I would argue that the paper would be made stronger by more accurately citing the relevant literature as to the origins of the probes and the fact that they have been successfully used in other labs.

D. Rather than referring to the free mCitrine as a "cell filler", the authors should probably refer to it as a "volume marker".

E. The authors refer to the activation of Rho following ectopic Rac activation as "strong" or "robust". It is not clear what is meant by either of these terms. In some of the data, the activation is apparently less than 10% above control levels, which doesn't seem particularly strong. And in other experiments (see, for example, Fig. 3C), there seems to be considerable variation in the amount of Rho activation achieved, ranging from a 10% increase to a 10% decrease, which is sort of the opposite of robust.

Reviewer #3 (Remarks to the Author):

The authors use perturbation of Rho GTPase location and biosensors for Rho GTPase activity to examine spatiotemporal aspects of Rho GTPase activity and crosstalk. A number of different cell lines (Neuro-2a, HeLa, U2OS, NIH3T3, A431) were used for these single cell studies. The perturbations are done with a chemically induced dimerisation system or with optogenetics. The biosensors were based on binding domains that have higher affinity for the active, GTP-bound Rho GTPase. New variants of these biosensors were generated and used. Using these tools, the effects of different GEFs on protrusion-retraction dynamics were studied.

The study touches on many different and interesting aspects of RhoGTPase activity. For instance, a surprising observation is that the recruitment of a constitutive active Rac leads to activation of Rho (figure 2C). However, many of the different observations are weakly connected. The evidence that Rac activates Rho through the ARHGEF11&12 is weak and indirect. Find below my specific comments and suggestions to improve the manuscript.

1. The measurements of activation and sensor response in figure 2C have different scaling of the axes. In particular for the sensor response. This makes it easy to compare kinetics, but hard to compare the amplitude of the sensor response. The amplitudes are shown in figure 2D, but in that panel the axes of the bar graphs are cut. I recommend to improve the display of these data. The authors may consider a constant scale across the figures in panel C.

Cutting axes in bar graphs is bad practice and therefore should be corrected for panel 2D. As an

alternative, a dot plot would be an option for panel D to show the data. I understand that 3 independent experiments were done, so actually $n=3$ and this can be better displayed with a superplot (<https://doi.org/10.1083/jcb.202001064>). The same is true for panel 2E, which can be replaced by a superplot.

2. Can the authors add a discussion of the kinetics of the recruitment+activity (panel 2C) which seems complex (biphasic at least) and different between experiments? Particularly the kinetics of recruitment of Cdc42 and RhoA when a biosensor for Rho is used stand out (upper right and lower right plot) from the other data.

The recruitment efficiency of Cdc42 is only half (upper row in panel 2C) compared to that of Rac and Rho. What is the reason (biological or technical)? Would that affect the interpretation?

3. In panel 2B the raw responses are shown for perturbation, sensor and control. The control shows a substantial decrease in signal, while the other two responses (Rac1 perturbation, Rho response) return to baseline levels around $t=40$ minutes. I do not see how the difference between the perturbation/sensor and the control can be so different? Did the authors look into photobleaching kinetics of the different components (in our hands CFP and RFP (mCherry) are far more photo stable than YFP)? Additionally, I do not see large morphological changes in Movie 2, and so it looks like the signal in the control is bleaching? Did the authors perform control experiments where no perturbation was done to check for bleaching or photochromism?

4. In our hands, the PA-Rac is tricky to work with, as it has substantial constitutive activity. Did the authors notice any basal activity (perhaps judged by cell size or morphological features)? Given the heterogeneity of responses (figure 3C), can the authors connect the differences in the Rho sensor responses to the expression level of PA-Rac (as quantified by the intensity of CFP)?

5. In the legend of figure 3, the authors mention that 67% of cells show a response and it seems that the subset of responding cells is used to generate panel D and E (but I might be wrong). Is that right? How is the heterogeneity for the other cell lines (panel 3B) and what percentage of cells did not respond? It may be interesting and more transparent to show the responses of all cells, for instance in a heatmap-style plot. (For examples see figure 1H of this paper: <https://www.embopress.org/doi/full/10.15252/msb.20156458>). This will also show all responses scaled in the same way right now, the graphs in panel C have different scaling, which makes it harder to compare the responses).

6. New biosensors were generated. The authors state "This alternative sensor design might lead to increased interactions with active GTPases due to increased avidity", and also "Using these improved sensors". There is, however, no validation or characterisation of the new sensors, so it is unclear whether the new sensors are an improvement. It is known that increasing the number of binding domains for Rho GTPases can increase avidity, but the extent is not linearly dependent on the number of domains and may also depend on architecture

(<https://journals.biologists.com/jcs/article/134/17/jcs258823/272101/Visualizing-endogenous-Rho-activity-with-an>). Therefore, validation of the new biosensors is key. A minimal piece of data is the comparison of affinity (measured as relocation efficiency) between the new and old sensors. In addition,

specificity for Rho GTPases is an important parameter that need to be addressed.

7. I recommend that the authors share the newly generated biosensors with the community by depositing the plasmids on addgene.

8. The data visualization in figure 4 and the corresponding movies are nice and convincing. I think it would be helpful to add a panel for Rho activity in relation to cell edge velocity (similar to panel 4C).

9. The authors observe that ARHGEF11 and ARHGEF12 are enriched at the cell edge. I think that the use of a membrane bound protein as a measure for increased accumulation of membrane is a better control here (see for instance figure 1: <https://www.sciencedirect.com/science/article/pii/S096098220201223X> or

Fig. 6: <https://doi.org/10.1242/jcs.258823>) instead of a cytoplasmic protein. A cytoplasmic protein does not reflect increased amounts of membrane, which would also result in an increase in intensity for a (partially) membrane associated protein.

10. The local enrichment of ARHGEF11&12 (if it is not due to local membrane accumulation, see point 9) is striking and exciting. Incidentally, both ARHGEF11&12 can be bound to and activated by Galpha12/13. This is, however, not mentioned or discussed although the literature is cited (Fukuhara et al. 1999, 2000) to state that these RhoGEFs activate Rho. The local enrichment of RhoGEFs may as well be triggered by Galpha12/13 (which would be equally interesting). Can the authors comment on the link between G12/13 activation and local enrichment of ARHGEF11&12? Can they exclude that G12/13 is responsible for the recruitment?

11. Overall, the evidence that the enrichment at the cell edge is due to increased Rac is weak and indirect. The experiment with PA-Rac (panel 5F) shows only a marginal increase in fluorescence after optogenetic activation. Still, if this is true, reaching maximal activation ~20s, why is there a substantial delay (and order of magnitude larger) between Rac activation and ARHGEF recruitment (160s-210s). In the previous experiments with PA-Rac, activating Rho (figure 3) there was an immediate effect. The kinetics do not seem to match.

12. The authors write "A previous biochemical study suggested that Rho activators of the Lbc GEF family might act as effectors of active Rac (Figure 5A) and thereby might mediate this activity crosstalk (Dada et al., 2017)". The paper by Dada et al. focusses on p190 and uses an in vitro approach. In that paper, the authors find a weak interaction between Rac1 and p190 and that "the binding affinity between the PH domain of p190 and activated RhoA is at least 10 fold higher than that towards activated Rac1". The p190GEF is also known as ARHGEF28, which the authors include in their localization studies, but it shows no accumulation at protrusions.

On the other hand, Dada et al write "... pulldown assays with activated Rac1 did not detect interactions with the PH domains of various Lbc RhoGEFs (PRG, LARG, and GEFH1)". So the biochemical data conflicts with the observations in this manuscript, rather than supporting them. This should be included in the discussion. Since a direct interaction between Rac and the ARHGEF11&12 is an important aspect of the study, the authors need to provide better and more convincing (direct) evidence.

13. The authors do knock-downs and overexpression of ARHGEF11 or ARHGEF12, which result in altered protrusion-contraction dynamics. These data are consistent with the proposed roles in the crosstalk, but they do not provide evidence for this model.

14. In the discussion, the authors write "several earlier studies suggested a potential activating role of Rac on Rho (Bustos et al., 2008; Guilluy et al., 2011; Nimnual et al., 2003; Rosenfeldt et al., 2006; Sander et al., 1999)". However, I read the opposite in most of the references:

Bustos et al 2008: "We further demonstrate that activated Rac1 depletes RhoA-GTP from membranes of HeLa cells"

Nimnual et al., 2003: "we have shown that ROS production is an essential component in the signalling cascade that mediates Rac-induced downregulation of Rho "

Rosenfeldt et al., 2006: "In this study, we show that the ability of thrombin to promote stress fiber formation and stimulate Rho can be inhibited by activated forms of Rac"

Sander et al., 1999: "Both sustained Cdc42 and Rac activation result in downregulation of endogenous Rho activity, demonstrating that Cdc42 and Rac antagonize Rho by regulating its GTP level"

The only support for the statement is in Guilluy et al., 2011. This review mentions one specific way in which Rac activates Rho: "Rac1-GTP binds to the PH domain of Dbs, a RhoA GEF [23,24], and stimulates its catalytic activity, leading to RhoA activation".

In the context of the current paper, this is a relevant connection between Rac and Rho and it should be mentioned and also discussed whether this can also occur in the cell systems that are studied here. Why did the authors not look into activation of Dbs by Rac?

15. The authors mention "acute perturbations" and I wondered what the meaning of acute is in this context. The best fitting synonyms that I could find are intense/sharp/drastic. Is this what the authors mean?

16. The authors observe 'formation of cell protrusions' for Cdc42 and Rac activation (figure 1). Do they notice a difference between the two GTPases. In their introduction they mention that Rac induces flat cell protrusions and Cdc42 pointed protrusions, so it is relevant to compare the morphological changes induced by Cdc42 and Rac.

Reviewed by Joachim Goedhart (University of Amsterdam)

Response to reviewers' comments:

We thank the reviewers for their constructive criticism which have allowed us to revise and improve our manuscript. A point-by-point discussion is presented below. Our responses are highlighted in blue:

Reviewer #1 (Remarks to the Author):

In this interesting study, the authors investigate the crosstalk between the GTPases, Rac and Rho. In the first part of the paper, the authors used and optimized two previously developed assays to study the crosstalk of Rho GTPases in N2A cells. Using these assays, they discover that artificial Rac activation causes Rho activation. In the second part of the paper, they use A431 cells to understand the mechanism of the regulation of Rho activation by Rac. Using a cross-correlation analysis approach, they find that Rac GTPase is activated before Rho GTPase during a protrusion event. They identify the RhoGEFs Arhgef11 and Arhgef12 as novel regulators of the Rac-mediated activation of Rho. They modulate the expression level of Arhgef11 and Arhgef12 by overexpression and knockdown and identify 2 phenotypes to explain the effect of Arhgef11 and Arhgef12 on the Rac-mediated activation of Rho.

Response: We thank the reviewer for these positive and supportive comments regarding our work.

Major points. The manuscript should in my view be published if the concerns below can be addressed:

1) The authors need to show data and discuss if there are spatial or temporal constraints for the mechanism of Rac to Rho activation they describe. For example, is the mechanism they describe only happening at the front of a migrating cell or also at the back of a migrating cell? Can the authors comment on how their assay is affected by measuring signal at the cell edge vs. the center of the cell?

Response: We thank the reviewer for pointing this out. Concerning the Rac-dependent Rho activation, we do not observe any significant spatial constrains in any of the cell lines that we tested and we now provide an additional analysis that measures the response either at cell attachment regions in the cell periphery or near the cell center and show that there is no difference in the Rac/Rho crosstalk. Concerning the response kinetics, we now included a more detailed discussion, also in response to more specific questions below (page 17, para 3). Concerning more detailed aspects of the crosstalk mechanism, we were able to detect some spatial constrains. In particular, the crosstalk mediators Arhgef11/12 are enriched in the cell periphery and measurements of their recruitment showed that their recruitment occurs preferentially near the cell center at early time points and preferentially near the periphery at later timepoints. We discuss these observations in the context of F-actin as a potential crosstalk mediator and included these new analyses in the revised manuscript.

2) In figure 2 and 3, the authors show that Rho is activated upon Rac stimulation. They need to show whether knockdown of Arhgef11 and Arhgef12 change the activation of Rho. I recommend that the authors use the dimerization-based or optogenetics-based approach and show in more detail the effect of Arhgef11 and Arhgef12 knockdowns on changes in the strength and duration of Rho activation.

Response: We thank the reviewer for this important comment. We now performed additional experiments in which we combined RNAi-mediated knockdown of Arhgef11 and Arhgef12 with optogenetic Rac1 perturbations. These experiments showed that both GEFs significantly contribute to Rac1-stimulated Rho activation, which further supports our findings (Figure 5 and S5).

3) In figure 6, the authors show some effect of Arhgef11 and Arhgef12 knockdowns on protrusion-retraction cycle duration. However, the effect of siRNA control on reducing protrusion-retraction cycle duration is as strong as the effect of the Arhgef11 or Arhgef12 overexpression. The authors need to include data using a WT control and discuss why siControl may have such an effect. Similarly, the authors should include a WT control for the data presented in figure 6f-g.

Response: We thank the reviewer for this important point. Indeed, the morphodynamic measurements for the control transfection are similar to the measurements after GEF knockdown. We would like to argue, that the two experimental conditions that are used in the control conditions are very different from each other. In particular, the cells used for the RNAi experiments were transfected twice with two different transfection reagents compared to the overexpression experiments. We therefore believe that comparisons should only be made within these two different sets of experiments.

4) To test if Arhgef11 and Arhgef12 work synergistically, the authors could do a double knockdown of Arhgef11 and Arhgef12.

Unfortunately, double knockdown of Arhgef11 and Arhgef12 using the conditions that we optimized for single knockdown strongly reduced the number of viable cells, presumably due to excessive cell death or reduced proliferation. We believe that understanding synergism between Arhgef11 and Arhgef12 would require substantial optimization and that this aspect of Rac/Rho crosstalk would be beyond the scope of this study.

Minor points

5) Figure 1 is presented in a confusing way as it introduces 2 fluorescent markers but then in 1B does not show how the signal of the reporter looks like but instead shows an Actin reporter. Please add the actual image data of the reporter in this figure and make clear which reporters are used.

Response: We initially focused on the actin reporter, which was used to investigate the morphometric changes independently of the altered signal of the Rac1/RhoA and Cdc42 perturbation constructs. We clarified this now in the revised figure 1 and added a new Figure S1 which shows the recruitment of the perturbation constructs.

6) In figure 2C, they should use the same scale for the control sensor corrected response signal in all plots.

Response: Also related to comment #1 from reviewer #3, we now use the same scale to represent this data. However, to point out the most important take home messages from this

data set, we divided the results between the main figure and a supplementary figure. In the main figure, we focus on the most significant crosstalk results, which we also comment on in the main text. In the supplementary figure, we show all the combinations, in particular to point out that the matching perturbation/sensor pairs in the diagonal of this matrix show the strongest response.

7) In figure 2, did the authors correct for possible FRET interaction of their BFP and YFP reporter?

Response: We do not believe that FRET would significantly affect our measurements. To avoid contaminating FRET signals, we used narrow emission filter sets and only excited one fluorophore at the time. We clarified this point in the methods section (page 29, para 1). Quenching due to FRET is also not expected: For RFP no acceptor is present. The YFP fluorophore is cytosolic and therefore is not expected to come into close proximity with the RFP or CFP fluorophore. Therefore, neither quenching of YFP, nor of CFP is expected. Quenching via FRET of BFP by CFP might indeed occur. However, the BFP signal intensity is not central to our measurements, as we used BFP in excess, and we only confirmed its expression in cells at the end of the measurements.

8) In figure 3, the authors use a different duration for the perturbation, 250s (4.17 min) instead of 20 min in figure 2. Can they comment on why they chose a different duration?

Response: It takes more time for the dimerizer to enter the cell compared to the immediate effect of switching on the optogenetic tool. We clarified this point in the methods section (page 30, para 2).

9) In figure 3, they only show the Rho activation data for the A431 cell line. However, all experiments so far have used the N2A cells. Can the authors include the Rho activity dynamics over time for these cells as well?

Response: Also related to the comment of reviewer #2, we added this data in Figure S3.

10) In figure 3C, there are 4 different scenarios of Rho activity, upon Rac activation, presented. The 3 lower panels (in sum 71% of the cells) on the right all have a downward trend of Rho activity from the pre-stimulus baseline signal after perturbation with Rac activity. Can the authors present the data using the same scale and comment on this effect?

Response: Indeed, in sum 71% of the cells show a negative trend at longer timescales. However, at shorter timescales, 54% of cells do show at least a transient increase. We adjusted the scale of all panels and also added a heatmap style representation of all responses using the same scale in response to reviewer #3. We also extended our discussion in the main text of potential negative feedback that might lead to this effect (page 10, para 2).

11) In figure 5e, the authors show that there is a temporal sequence of events with Rac peaking first, then Arhgef11 and Arhgef12, then Rho. However, Rho activity is also decreasing while Rac and Arhgef11 and Arhgef12 are already increasing. Can the authors comment on the decline of Rho activity during that period?

Response: This is indeed a very interesting point. The reciprocal crosstalk between Rac and Rho might be responsible for this effect. Based on previous studies and on our crosstalk analysis shown in Figure 2, active Rho can inhibit Rac activity. Thus, Rho would need to be inhibited to enable triggering of a new protrusion. This could for example be mediated by negative feedback. We included these points in the discussion section (page 20 para 4).

12) Also, the section in which Rho is reaching its peak is cut off from the figure. It seems that Rho activity is still increasing.

Response: We extended the range of data shown in these plots, which now also includes the datapoints at which Rho is maximal after protrusion.

13) What is the reasoning for the time scale of Rac-dependent activation of Rho? Can the authors give discuss why the Rac-dependent activation of Rho may last for +20 min while the inhibitory effect of Rho on Rac is only apparent at 5 min after Rho activation?

Response: We thank the reviewer for this important comment. We believe that the differences in the response kinetics are due to differences in feedback mechanisms that lead to adaptation during the perturbation. Interestingly, the perturbations are based on dominant positive mutants, which cannot be inhibited themselves. Therefore, the adaptation has to occur downstream in the crosstalk mechanism. Furthermore, this adaptation is not only specific to the type of crosstalk (Rho activation by Rac vs Rac inhibition by Rho), but also cell-type specific: While the Rho response to a Rac1 perturbation continues to rise in N2a cells over a prolonged time period, the corresponding response quickly adapts in A431 cells. In our revised manuscript we propose a mechanism that can lead to adaptation of the Rac-triggered Rho response based on myosin-mediated Rho self-inhibition (page 17, para 3), which might involve cell-type specific components. We also argue that the Rho-induced Rac1 inhibition might involve similar adaptive feedback mechanisms.

14) The authors should use standard deviation for indicating the variability of their data, in addition/instead of standard error of the mean throughout the paper.

Response: Due to the high level of cell-to-cell variability inherent to our experimental system, we would argue that reporting the precision of our average measurements is most relevant for our conclusions and we therefore prefer to use the standard error of the mean for the majority of the panels in the main manuscript. However, we also agree that it is important to show the variability of the data. Therefore, and also in response to reviewer #3, we now use dot plots instead of bar graphs to visualize the variability of our measurements.

Reviewer #2 (Remarks to the Author):

Review of “Crosstalk between Rac and Rho GTPase activity mediated by Arhgef11 and Arhgef12 coordinates cell protrusion-retraction cycles” by Nanda et al.

The cell cortex is patterned by Rho GTPases during a variety of fundamental biological processes such as cell locomotion, cell division, and cell repair. In many situations, crosstalk between the three most-studied Rho GTPases—Rho, Rac, and Cdc42—is thought to contribute to such cortical patterns.

In the current study, the authors address cortical patterning during the protrusion-retraction cycle of cell locomotion. To track Rho GTPase dynamics, the authors used probes based on diffusion trapping, in which soluble, Fluorescent protein (FP) GTPase activity reporters are recruited from a cytoplasmic pool to the plasma membrane at sites of GTPase activation. This is not a new approach (see also below) as it was developed nearly 20 years ago for studies of cell repair and cytokinesis. However, for reasons that are not at all clear, studies of cell locomotion have usually relied on FRET-based reporters which, strictly speaking, do not measure endogenous GTPase activity.

The authors first show that experimental Rac activation by two different approaches induces Rho activation in several different cell lines. This is both a surprising and an exciting finding. It’s surprising in that it has generally been thought that Rac and Rho are mutually antagonistic during cell locomotion. It’s exciting because it suggests a simple (on paper, anyway) link between protrusion and retraction: Rac activation first leads to protrusion but then, due to (presumably) a delayed activation of Rho, retraction will naturally ensue.

The authors then image Rac and Rho activity during natural cell locomotion. Consistent with the above findings, they see that cortical pulses of Rac activity immediately precede protrusion while similar pulses of Rho activity immediately precede retraction. Finally, the authors identify two GEFs, Arhgef11 and Arhgef12, as potential links between Rac and Rho.

This is important work that will be of considerable interest to students of Rho GTPase signaling and cell locomotion. For the former, the results challenge a treasured dogma—that Rac antagonizes Rho and vice versa. Similarly, the successful deployment of the simple GBD-based probes to study cell motility may change the impression that such probes are inherently inferior to FRET-based probes. For the latter, the model developed by the authors provides an elegant, plausible explanation for how protrusion can be coupled to retraction.

Response: We thank the reviewer for the positive review of your work and its impact on the Rho GTPase community.

I have no major concerns with the study, but there are several areas that could do with some revision or further explanation. These are presented below:

A. While it is clear that ectopic activation of Rac induces Rho activation in most cell types, it was not so clear whether there is a consistent delay between the two and for how long the delay lasts. Presumably, the effect cannot be instantaneous, but I was simply unable to determine this from the data presented.

Response: Related to a comment of reviewer #1, we added the averaged response curves for all cell lines in the supplementary material (Figure S3). These measurements show that the most striking differences are in the extent of the response and in the adaptation during optogenetic perturbation. The initial response to the optogenetic perturbation is consistently very fast, presumably due to efficient catalytic activation of Rho by the recruitment of the Rho GEFs.

B. The authors have done a nice job revealing Rho pulses following Rac activation; one cannot help but wonder if depletion of Arhgef11 or Arhgef12 (or both) reduces the amplitude of these pulses. However, I recognize this is an easy experiment to request on paper but may be quite challenging in practice, as periodic behaviors such as the authors are studying can vary considerably under control conditions.

Response: Related to comment #3 of reviewer #1, we combined Arhgef11/12 depletion with measurements of the Rho activity response to optogenetic Rac1 perturbations. These experiments clearly showed that both Arhgef11 and Arhgef12 are required for the Rho response after rapid Rac1 stimulation (Figure 5 and S5).

C. Based on the abstract and the introduction, I had the impression that the authors developed improved Rho activity sensors in this paper. However, upon closer reading of the manuscript, I learned otherwise: The authors state that "...we used TIRF microscopy-based sensor constructs that we developed previously (Graessl et al., 2017)". I took this to mean that the probes are somehow specifically useful for TIRF and that they were developed by the authors. However, upon checking the Graessl paper, it turns out that the Rho and Cdc42 probes were simply FPs fused to Rhotekin rGBD and NWASP GBD, respectively. The FP-rGBD approach was developed for live cell imaging in 2005 as described in PMID: 15684032. The FP-wGBD probe approach was developed for live imaging in 2003 by two different labs more or less simultaneously (PMIDs: 12874226 and 12872130). These probes do indeed work with TIRF but they also work with a variety of other imaging modalities and as far as I can tell they were not modified for TIRF in either Graessl et al. or in the current study.

Later, the authors state that they increased the sensitivity of the probes by generating "...constructs that contain tandem GTPase binding domains that could benefit from binding multiple GTPase molecules at the same time. This alternative sensor design might lead to increased interactions with active GTPases due to increased avidity" and then "Using these improved sensors...". The implications here are that the authors show that these sensors are more sensitive and, given the context, that this is something that hasn't been tried before. However, nowhere in the paper is there any kind of comparison, quantitative or otherwise, of the original sensors (based on a single GBD) to the "improved" sensors (based on two GBDs). With respect to novelty, the authors use mCherry-2X-rGBD as an improved sensor for active Rho. This exact construct was used in 2016 (PMID: 27226483). With respect to demonstration of improvement, while not in the current manuscript, a systematic study from 2021 showed introduction of two or more GBDs into Rho biosensor probes improves their sensitivity (PMID: 34357388).

This is not to say that the choice of probes doesn't matter because it does, as the authors make clear in the Discussion. Nor is it to say that this paper is somehow less noteworthy

because the probe technology is not new. Indeed, I would argue that the paper would be made stronger by more accurately citing the relevant literature as to the origins of the probes and the fact that they have been successfully used in other labs.

Response: We thank the reviewer for this important point. Indeed, the general concept of using translocation sensors to measure Rho GTPase activity in cells is not new and we clarified this in the revised manuscript (page 8, para 1). However, for this study we generated new plasmid constructs and combined multiple aspects to improve our experiments that collectively were critical for our investigations of migrating cells: 1) Avidity (multiple GBDs), 2) Sensitivity (TIRF microscopy), 3) low expression (delCMV promotor). While each of these aspects was used earlier in published work, to the best of our knowledge, these three aspects were not combined in previous studies. To clarify these points, we cited additional studies (PMIDs: 15684032, 12874226, 12872130, 27226483, 34357388 and 37226883) and we added analyses that compare the newly generated delCMV-multi-GBD with the previous delCMV-single-GBD constructs (Figure S3).

D. Rather than referring to the free mCitrine as a “cell filler”, the authors should probably refer to it as a “volume marker”.

Response: We changed the naming as suggested to clarify its function.

E. The authors refer to the activation of Rho following ectopic Rac activation as “strong” or “robust”. It is not clear what is meant by either of these terms. In some of the data, the activation is apparently less than 10% above control levels, which doesn’t seem particularly strong. And in other experiments (see, for example, Fig. 3C), there seems to be considerable variation in the amount of Rho activation achieved, ranging from a 10% increase to a 10% decrease, which is sort of the opposite of robust.

Response: Our intention for this wording was to contrast the Rho response in N2a, HeLa and NIH3T3 cells, which indeed was highly reproducible, to the observations in U2OS and A431 cells, which was on average significant but indeed more varied and not robust. As this reviewer notes, our wording did not properly reflect this point. We revised the text and removed the general and unspecific statements “strong” and “robust” and we only kept one statement in which we compared different conditions, and where we indeed observed “stronger” and “more robust” responses (page 9, para 2).

Reviewer #3 (Remarks to the Author):

The authors use perturbation of Rho GTPase location and biosensors for Rho GTPase activity to examine spatiotemporal aspects of Rho GTPase activity and crosstalk. A number of different cell lines (Neuro-2a, HeLa, U2OS, NIH3T3, A431) were used for these single cell studies. The perturbations are done with a chemically induced dimerisation system or with optogenetics. The biosensors were based on binding domains that have higher affinity for the active, GTP-bound Rho GTPase. New variants of these biosensors were generated and used. Using these tools, the effects of different GEFs on protrusion-retraction dynamics were studied.

The study touches on many different and interesting aspects of RhoGTPase activity. For

instance, a surprising observation is that the recruitment of a constitutive active Rac leads to activation of Rho (figure 2C).

Response: We thank the reviewer for these positive comments on our study.

However, many of the different observations are weakly connected. The evidence that Rac activates Rho through the ARHGEF11&12 is weak and indirect. Find below my specific comments and suggestions to improve the manuscript.

Response: Based on the specific comments #12 and #13 by this reviewer, and comments from reviewer #1, we added additional experiments that further strengthen the role of ARHGEF11/12 in the Rac/Rho activity crosstalk (see replies to #12 and #13 below). We thank the reviewers for these very useful comments.

1. The measurements of activation and sensor response in figure 2C have different scaling of the axes. In particular for the sensor response. This makes it easy to compare kinetics, but hard to compare the amplitude of the sensor response. The amplitudes are shown in figure 2D, but in that panel the axes of the bar graphs are cut. I recommend to improve the display of these data. The authors may consider a constant scale across the figures in panel C. Cutting axes in bar graphs is bad practice and therefore should be corrected for panel 2D. As an alternative, a dot plot would be an option for panel D to show the data. I understand that 3 independent experiments were done, so actually $n=3$ and this can be better displayed with a superplot (<https://doi.org/10.1083/jcb.202001064>). The same is true for panel 2E, which can be replaced by a superplot.

Also in response to comment 14 of reviewer #1, we now use equal scales for all plots that are shown together in one figure. In the main figure we focus on the crosstalks that we particularly discuss in the main text and that are most important for this manuscript. Thereby, we make it easy to compare the kinetics in these conditions. In figure S2, we show all conditions to make it easy to compare the amplitude for all crosstalk combinations. Also, as suggested by this reviewer we now use dot plots in the new panels figure 2e and figure S2b. In addition, we also replaced all bar graphs with dot plots throughout the whole manuscript

2. Can the authors add a discussion of the kinetics of the recruitment+activity (panel 2C) which seems complex (biphasic at least) and different between experiments? Particularly the kinetics of recruitment of Cdc42 and RhoA when a biosensor for Rho is used stand out (upper right and lower right plot) from the other data. The recruitment efficiency of Cdc42 is only half (upper row in panel 2C) compared to that of Rac and Rho. What is the reason (biological or technical)? Would that affect the interpretation?

Response: As these experiments are based on addition of small molecules, we believe that several factors might affect the recruitment and responses kinetics, both technical and biological. In our response to comment #13 of reviewer #1 we already commented on apparent adaptation to the perturbations, which can occur at different levels of the signal network and we now discuss, how this could lead to some of the observed complexity. The perturbation kinetics in most cells corresponds to a biphasic kinetics, with a rapid first phase and slower second phase. These different phases could potentially be explained by simultaneously occurring processes of plasma membrane targeting and intracellular diffusion of the GTPases and uptake of the small molecule dimerizer through the plasma membrane.

As transient transfections were used for these experiments, differences in expression level might also account for some of the variability in the recruitment efficiency and kinetics.

The weaker perturbation kinetics of the Cdc42 construct is paired with a particularly strong response of the corresponding Cdc42 sensor. Therefore, although the amplitude of this perturbation is weaker compared to the Rac1 or RhoA perturbation, our data suggests that the resulting Cdc42 activity perturbation is equal or stronger and we therefore do not believe that we significantly underestimate the crosstalk in these conditions relative to other crosstalk measurements.

We added a brief discussion of these points in the main text of the revised manuscript (page 8, para 3). For the remainder of the manuscript, we particularly focused on the Rac/Rho crosstalk, which was most interesting to us, and confirmed this observation with an independent, optogenetic method. We believe that a more detailed investigation into the other crosstalk kinetics is beyond the scope of the current study.

3. In panel 2B the raw responses are shown for perturbation, sensor and control. The control shows a substantial decrease in signal, while the other two responses (Rac1 perturbation, Rho response) return to baseline levels around t=40 minutes. I do not see how the difference between the perturbation/sensor and the control can be so different? Did the authors look into photobleaching kinetics of the different components (in our hands CFP and RFP (mCherry) are far more photo stable than YFP)? Additionally, I do not see large morphological changes in Movie 2, and so it looks like the signal in the control is bleaching? Did the authors perform control experiments where no perturbation was done to check for bleaching or photochromism?

Response: During TIRF microscopy, we do not observe significant photobleaching of cytosolic proteins, as only a small fraction of the cells is illuminated in the evanescent field. Furthermore, by using a sensitive EM-CCD camera, we were able to acquire these image series with little excitation intensity. Finally, a reduction in fluorescence intensity is only observed for mCitrine, which is particularly stable. We were also puzzled by this observation, and suspected that morphological changes outside the TIRF field might account of this effect. Indeed, we were able to detect an increase in cell volume after acute Rac1 perturbation, which might account for the reduced average intensity at the cell attachment area. We added this data in figure S1.

4. In our hands, the PA-Rac is tricky to work with, as it has substantial constitutive activity. Did the authors notice any basal activity (perhaps judged by cell size or morphological features)? Given the heterogeneity of responses (figure 3C), can the authors connect the differences in the Rho sensor responses to the expression level of PA-Rac (as quantified by the intensity of CFP)?

Response: We also observed some minor changes in cell morphology that presumably is due to basal activity of PA-Rac1. As suggested by this reviewer, we now include an additional plot in the supplementary materials that indeed shows correlation between PA-Rac1 level and the Rho response, showing that the crosstalk is dose-dependent.

5. In the legend of figure 3, the authors mention that 67% of cells show a response and it seems that the subset of responding cells is used to generate panel D and E (but I might be wrong). Is that right? How is the heterogeneity for the other cell lines (panel 3B) and what percentage of cells did not respond?

Response: The original Panels D and E (now Figure 3e and f) include all the cells – the same is true for panel B. In addition, the corresponding figure legend was confusing with regard to the 25% of cells that do not respond. We clarified these points in a revised figure legend. We also now show the responses of all cells via heatmaps (see below).

It may be interesting and more transparent to show the responses of all cells, for instance in a heatmap-style plot. (For examples see figure 1H of this paper: <https://www.embopress.org/doi/full/10.15252/msb.20156458>). This will also show all responses scaled in the same way right now, the graphs in panel C have different scaling, which makes it harder to compare the responses).

Response: We thank the reviewer for this helpful comment and now show the responses of all individual A431 cells in the main figure 3 and of all other cell types in Figure S3.

6. New biosensors were generated. The authors state "This alternative sensor design might lead to increased interactions with active GTPases due to increased avidity", and also "Using these improved sensors". There is, however, no validation or characterisation of the new sensors, so it is unclear whether the new sensors are an improvement. It is known that increasing the number of binding domains for Rho GTPases can increase avidity, but the extent is not linearly dependent on the number of domains and may also depend on architecture (<https://journals.biologists.com/jcs/article/134/17/jcs258823/272101/Visualizing-endogenous-Rho-activity-with-an>). Therefore, validation of the new biosensors is key. A minimal piece of data is the comparison of affinity (measured as relocation efficiency) between the new and old sensors. In addition, specificity for Rho GTPases is an important parameter that needs to be addressed.

Response: Concerning specificity, we show in Figure S2b that the single domain sensors each correspond much stronger to the corresponding perturbation (diagonal vs off-diagonal responses), which further strengthens previous reports on the specificity of the used effector domains (page 9, para 2). Concerning efficiency, we now added additional data for recruitment of the Rac sensor to PA-Rac, and additional data for the recruitment of the rho sensor to nocodazole-stimulated Rho activity pulses (Figure S3 and page 11, para 3).

7. I recommend that the authors share the newly generated biosensors with the community by depositing the plasmids on Addgene.

Response: We agree with this reviewer and will share the constructs via Addgene as soon as the paper is accepted for publication.

8. The data visualization in figure 4 and the corresponding movies are nice and convincing. I

think it would be helpful to add a panel for Rho activity in relation to cell edge velocity (similar to panel 4C).

Response: We added this additional analysis in the supplementary materials (Figure S4).

9. The authors observe that ARHGEF11 and ARHGEF12 are enriched at the cell edge. I think that the use of a membrane bound protein as a measure for increased accumulation of membrane is a better control here (see for instance figure 1: <https://www.sciencedirect.com/science/article/pii/S096098220201223X> or Fig. 6: <https://doi.org/10.1242/jcs.258823>) instead of a cytoplasmic protein. A cytoplasmic protein does not reflect increased amounts of membrane, which would also result in an increase in intensity for a (partially) membrane associated protein.

Response: We added an additional analysis of Arhgef11 and Arhgef12 enrichment based on a CAAX-box membrane marker in Figure S5 (page 14, para 2). This additional analysis confirms our previous results. We prefer to keep the original analysis in the main figures, as we believe that the cytosolic marker is better suited: First, based on wide-field microscopy, the Arhgef11/12 fusion proteins are predominantly cytosolic and only a small fraction is localized at the plasma membrane. The same applies to the GBD-based activity sensors. Second, the CAAX-box based membrane anchor exchanges relatively slowly with the cytosol and therefore can lead to inhomogenous bleaching in peripheral vs central cell attachment areas during TIRF imaging, which could lead to an incorrect bias in the quantification of protein localization.

10. The local enrichment of ARHGEF11&12 (if it is not due to local membrane accumulation, see point 9) is striking and exciting. Incidentally, both ARHGEF11&12 can be bound to and activated by G12/13. This is, however, not mentioned or discussed although the literature is cited (Fukuhara et al. 1999, 2000) to state that these RhoGEFs activate Rho. The local enrichment of RhoGEFs may as well be triggered by G12/13 (which would be equally interesting). Can the authors comment on the link between G12/13 activation and local enrichment of ARHGEF11&12? Can they exclude that G12/13 is responsible for the recruitment?

Response: In our revised manuscript, we now discuss several potential mechanisms, that might be involved in Rac1-activity dependent plasma membrane recruitment of Arhgef11 and or Arhgef12, including direct interaction with active Rac1 via the PH domain, interactions with G12/13, proposed heterodimerization between Arhgef11/12, and a proposed F-actin-binding site on Arhgef11 (page 15, para 2). In addition, and in response to comments #12 and #13, we also performed experiments to more directly investigate the mechanism of Arhgef11/12 plasma membrane targeting by active Rac1 (see below).

11. Overall, the evidence that the enrichment at the cell edge is due to increased Rac is weak and indirect. The experiment with PA-Rac (panel 5F) shows only a marginal increase in fluorescence after optogenetic activation. Still, if this is true, reaching maximal activation ~20s, why is there a substantial delay (and order of magnitude larger) between Rac activation and ARHGEF recruitment (160s-210s). In the previous experiments with PA-Rac, activating Rho (figure 3) there was an immediate effect. The kinetics do not seem to match.

Response: Several factors could play a role in the observed kinetics. For example, due to the enzymatic activity of the GEF, Rho activity could be increased very efficiently. Also, positive feedback of Rho activity could further amplify Rho activity, and in combination with subsequent negative feedback could explain the rapid, transient, pulse like Rho response that we often observe in our experiments. We added a discussion of these kinetics in the revised text (page 20, para 2). To gain more direct evidence to support a role for ARHGEF11/12 in mediating the Rac/Rho crosstalk, and also in response to a similar comment #2 from reviewer 1, we now use RNAi to show that both GEFs significantly contribute to the activation of Rho after optogenetic activation of Rac (Figure 5 and Figure S5, page 16, para 2).

12. The authors write "A previous biochemical study suggested that Rho activators of the Lbc GEF family might act as effectors of active Rac (Figure 5A) and thereby might mediate this activity crosstalk (Dada et al., 2017)". The paper by Dada et al. focusses on p190 and uses an in vitro approach. In that paper, the authors find a weak interaction between Rac1 and p190 and that "the binding affinity between the PH domain of p190 and activated RhoA is at least 10 fold higher than that towards activated Rac1". The p190GEF is also known as ARHGEF28, which the authors include in their localization studies, but it shows no accumulation at protrusions.

On the other hand, Dada et al write "... pulldown assays with activated Rac1 did not detect interactions with the PH domains of various Lbc RhoGEFs (PRG, LARG, and GEFH1)". So the biochemical data conflicts with the observations in this manuscript, rather than supporting them. This should be included in the discussion. Since a direct interaction between Rac and the ARHGEF11&12 is an important aspect of the study, the authors need to provide better and more convincing (direct) evidence.

Response: This is a very helpful comment that has stimulated us to perform additional experiments. To directly investigate a role for the Arhgef11/12 PH domain in mediating the Rac/Rho crosstalk, we introduced point mutations in the PH domain that were previously shown to be responsible for the interaction between Lbc type GEFs and active Rac1. We then measured the plasma membrane recruitment of the mutant ARHGEF11/12 constructs after optogenetic Rac activation and found that plasma membrane recruitment of Arhgef12 is indeed strongly dependent on an intact PH domain. For Arhgef11, the PH domain mutations did not have an effect, and we therefore reasoned that this molecule might be recruited by a different mechanism. Indeed, we found that the actin-binding domain of Arhgef11 is required for efficient plasma membrane targeting of Arhgef11 downstream of active Rac1. We added new figure panels in Figure 5 and new text (starting at page 15, para 2) to discuss these results.

13. The authors do knock-downs and overexpression of ARHGEF11 or ARHGEF12, which result in altered protrusion-contraction dynamics. These data are consistent with the proposed roles in the crosstalk, but they do not provide evidence for this model.

Response: As noted in the response to comment #11 above, we now provide more direct evidence to support our model, by combining RNAi mediated depletion of Arhgef11/12 with measurements of Rho activation after optogenetic Rac stimulation (Figure 3 and Figure S3).

14. In the discussion, the authors write "several earlier studies suggested a potential activating role of Rac on Rho (Bustos et al., 2008; Guilluy et al., 2011; Nimmual et al., 2003;

Rosenfeldt et al., 2006; Sander et al., 1999)". However, I read the opposite in most of the references:

Bustos et al 2008: "We further demonstrate that activated Rac1 depletes RhoA-GTP from membranes of HeLa cells"

Nimnual et al., 2003: "we have shown that ROS production is an essential component in the signalling cascade that mediates Rac-induced downregulation of Rho "

Rosenfeldt et al., 2006: "In this study, we show that the ability of thrombin to promote stress fiber formation and stimulate Rho can be inhibited by activated forms of Rac"

Sander et al., 1999: "Both sustained Cdc42 and Rac activation result in downregulation of endogenous Rho activity, demonstrating that Cdc42 and Rac antagonize Rho by regulating its GTP level"

The only support for the statement is in Guilluy et al., 2011. This review mentions one specific way in which Rac activates Rho: "Rac1-GTP binds to the PH domain of Dbs, a RhoA GEF [23,24], and stimulates its catalytic activity, leading to RhoA activation".

In the context of the current paper, this is a relevant connection between Rac and Rho and it should be mentioned and also discussed whether this can also occur in the cell systems that are studied here. Why did the authors not look into activation of Dbs by Rac?

Response: We apologize for confusing way how we cited these earlier studies. The majority of these citations were grouped together at the end of the sentence, however, most of these references only applied to the first part of the sentence, in which we state that the concept of mutual inhibition between Rac and Rho is frequently cited. We now moved all these citations to the first part of the sentence to clarify this point.

Concerning the molecular mechanism of Rac/Rho crosstalk, we did not consider Dbs a high priority candidate, as the wild type molecule is primarily localized at the Golgi apparatus (PMID: 15531584) and we did not consider this molecule as a good candidate for the very rapid Rac/Rho crosstalk that we observed in our study. The work that was cited in Guilluy et al used a truncated, oncogenic form of Dbs that was missing the domain that is responsible for Golgi targeting.

15. The authors mention "acute perturbations" and I wondered what the meaning of acute is in this context. The best fitting synonyms that I could find are intense/sharp/drastring. Is this what the authors mean?

Response: We apologize for this inaccurate wording. We now used the word "rapid" instead.

16. The authors observe 'formation of cell protrusions' for Cdc42 and Rac activation (figure 1). Do they notice a difference between the two GTPases. In their introduction they mention that Rac induces flat cell protrusions and Cdc42 pointed protrusions, so it is relevant to compare the morphological changes induced by Cdc42 and Rac.

Response: In the N2a cell system, Cdc42 induced protrusions that included both lamellipodia and filopodia. We revised our paper accordingly and report this observation in the results section (page 7, para 1). We also cited a paper that reports on the role of Cdc42 in stimulating lamellipodia, and we reduced the strength of our statement in the introduction.

Reviewed by Joachim Goedhart (University of Amsterdam)

REVIEWERS' COMMENTS

Reviewer #1 (Remarks to the Author):

my concerns have been addressed. I believe the manuscript is now ready for publication.

Reviewer #2 (Remarks to the Author):

The authors have carefully and thoroughly addressed all of my concerns. I look forward to the publication of this exciting work.

Reviewer #3 (Remarks to the Author):

The authors have taken all comments seriously and provide adequate responses. They have done a great effort, resulting in a substantially improved manuscript.

I agree that the new data with ARHGEF11/12 PH domain mutants strengthens the notion that these GEFs are involved in the Rac-mediated Rho activation, which is an exciting finding.

Reviewed by Joachim Goedhart (University of Amsterdam)